# LAP2alpha maintains a mobile and low assembly state of A-type lamins in the nuclear interior

Nana Naetar[1†]*, Konstantina Georgiou[1†], Christian Knapp[1‡], Irena Bronshtein[2], Elisabeth Zier[1], Petra Fichtinger[1], Thomas Dechat[1§], Yuval Garini[2#], Roland Foisner[1]*

[1]Max Perutz Labs, Center for Medical Biochemistry, Medical University of Vienna, Vienna Biocenter Campus (VBC), Vienna, Austria; [2]Physics Department and Nanotechnology Institute, Bar Ilan University, Ramat Gan, Israel

*For correspondence:
nana.naetar@univie.ac.at (NN);
roland.foisner@meduniwien.ac.at
(RF)

†These authors contributed
equally to this work

Present address: ‡ICFO – Institut
de Ciencies Fotoniques, The
Barcelona Institute of Science
and Technology, Castelldefels,
Spain; §Ludwig Boltzmann
Institute of Osteology at
Hanusch Hospital of OEGK and
AUVA Trauma Centre Meidling,
1(st) Medical Department
Hanusch Hospital, Vienna,
Austria; #Biomedical Engineering
Department, Technion - Israel
Institute of Technology, Haifa,
Israel

Competing interests: The
authors declare that no
competing interests exist.

Reviewing editor: Megan C
King, Yale School of Medicine,
United States

**Abstract** Lamins form stable filaments at the nuclear periphery in metazoans. Unlike B-type lamins, lamins A and C localize also in the nuclear interior, where they interact with lamin-associated polypeptide 2 alpha (LAP2α). Using antibody labeling, we previously observed a depletion of nucleoplasmic A-type lamins in mouse cells lacking LAP2α. Here, we show that loss of LAP2α actually causes formation of larger, biochemically stable lamin A/C structures in the nuclear interior that are inaccessible to lamin A/C antibodies. While nucleoplasmic lamin A forms from newly expressed pre-lamin A during processing and from soluble mitotic lamins in a LAP2α-independent manner, binding of LAP2α to lamin A/C during interphase inhibits formation of higher order structures, keeping nucleoplasmic lamin A/C in a mobile state independent of lamin A/C S22 phosphorylation. We propose that LAP2α is essential to maintain a mobile lamin A/C pool in the nuclear interior, which is required for proper nuclear functions.

## Introduction

Lamins are intermediate filament proteins in metazoan nuclei that, together with numerous inner nuclear membrane proteins, form a filamentous protein meshwork at the nuclear periphery, called the nuclear lamina (*Gruenbaum and Foisner, 2015a*). Based on their biochemical properties and expression patterns, lamins are grouped into two subtypes, A-type and B-type lamins. B-type lamins are ubiquitously expressed in all cell types and throughout development (*Yang et al., 2011*), whereas A-type lamins are expressed at low levels in embryonic stem cells and undifferentiated cells, but are significantly upregulated during differentiation (*Constantinescu et al., 2006*; *Eckersley-Maslin et al., 2013*; *Röber et al., 1989*). In mammalian cells, the major A-type lamins are lamins A and C encoded by the *Lmna* gene, whereas the two major B-type lamins, lamins B1 and B2, are encoded by *Lmnb1* and *Lmnb2*, respectively (*Gruenbaum and Foisner, 2015a*). Both lamin subtypes share a similar intermediate filament protein-type domain structure with a central rod domain, an N-terminal head and a globular C-terminal tail, which contains a nuclear localization signal, an Ig fold and, except for lamin C, a C-terminal CaaX motif (C: cysteine; a: aliphatic amino acid; X: any amino acid) (*de Leeuw et al., 2018*; *Gruenbaum and Medalia, 2015b*). The CaaX motif undergoes a series of post-translational modifications, including farnesylation of the cysteine, removal of the last three amino acids, followed by carboxymethylation (*Rusiñol and Sinensky, 2006*). Whereas B-type lamins remain farnesylated and carboxymethylated, pre-lamin A undergoes an additional processing step catalyzed by the metalloprotease Zmpste24, leading to the removal of 15 amino acids from its C-terminus, including the farnesylated cysteine residue. As a consequence, mature B-type lamins are tightly associated with the nuclear membrane via their farnesylated C-terminus and mainly localize at

the nuclear periphery, whereas mature lamin A is found both in filaments within the peripheral nuclear lamina, and additionally in a soluble and dynamic pool in the nuclear interior (*Naetar et al., 2017*). Lamin C that lacks a CaaX motif contributes also to the peripheral and nucleoplasmic pool of A-type lamins.

Lamin filaments at the nuclear periphery were recently visualized by cryo-electron tomography (*Turgay et al., 2017*), but their structure and assembly are far from being well understood. Lamins form dimers via their central rod domains, which further assemble into head-to-tail polymers (*de Leeuw et al., 2018*). Recent work by the Medalia lab using cryo-electron tomography revealed that in mammalian nuclei two head-to-tail filaments assemble laterally into 3.5-nm-thick filaments in a staggered fashion (*Turgay et al., 2017*). Lamin filaments at the nuclear periphery are considered stable, resistant to biochemical extraction and highly immobile (*Bronshtein et al., 2015*; *Moir et al., 2000*; *Shimi et al., 2008*). Together with proteins of the inner nuclear membrane, they fulfill essential functions, defining the mechanical properties of nuclei (*Buxboim et al., 2014*; *Cho et al., 2019*; *Davidson and Lammerding, 2014*; *Osmanagic-Myers et al., 2015*; *Swift et al., 2013*) and regulating higher order chromatin organization through anchorage of peripheral heterochromatic genomic regions (*Solovei et al., 2013*; *van Steensel and Belmont, 2017*).

The regulation and properties of A-type lamins in the nuclear interior are far less understood, although recent studies have unraveled novel functions of lamins in the nuclear interior in chromatin regulation that are fundamentally different from those fulfilled by the nuclear lamina (*Gesson et al., 2014*; *Naetar et al., 2017*). Nucleoplasmic lamins A and C bind to and regulate euchromatic regions of the genome, globally affecting epigenetic modifications and possibly chromatin accessibility (*Gesson et al., 2016*; *Naetar et al., 2017*). They also provide chromatin 'docking sites' in the nucleoplasm slowing down chromatin movement in nuclear space (*Bronshtein et al., 2015*). Moreover, intranuclear A-type lamins are required for the proper assembly of repressive polycomb protein foci (*Bianchi et al., 2020*; *Cesarini et al., 2015*) and are involved in the regulation of telomere function (*Chojnowski et al., 2015*; *Gonzalez-Suarez et al., 2009*; *Wood et al., 2014*). Nucleoplasmic lamins were also found to affect gene expression directly by binding to gene regulatory sequences, such as promoters and enhancers (*Briand et al., 2018*; *Ikegami et al., 2020*; *Oldenburg et al., 2017*).

The structure and assembly state of nucleoplasmic lamins remain enigmatic. Fluorescence recovery after photobleaching (FRAP), continuous photobleaching (CP), and fluorescence correlation spectroscopy (FCS) studies of fluorescently tagged A-type lamins showed that the majority of nucleoplasmic lamin complexes are highly mobile compared to the stable peripheral lamina (*Bronshtein et al., 2015*; *Moir et al., 2000*; *Shimi et al., 2008*). Biochemical studies revealed that intranuclear lamins can be easily extracted in salt and detergent-containing buffers, suggesting that they have a low assembly state (*Kolb et al., 2011*; *Naetar et al., 2008*) and/or weaker interactions with nuclear protein complexes and chromatin compared to the lamins at the lamina. Taken together, the dynamic nucleoplasmic lamin pool has very distinct properties compared to the static and stable peripheral lamina, allowing them to fulfill a unique set of functions. However, why and how nucleoplasmic A-type lamins display such different properties in the nuclear interior compared to their counterparts at the nuclear lamina remains poorly understood. Possible mechanisms include post-translational modifications and/or interactions with specific nucleoplasmic binding partners. For example, phosphorylation of lamins influences the localization, solubility and mobility of lamins both during mitosis and interphase (*Kochin et al., 2014*; *Machowska et al., 2015*). Moreover, the specific interaction partner of nucleoplasmic lamins, lamin-associated polypeptide 2 alpha (LAP2α) was suggested to regulate the localization of lamin A/C in the nuclear interior, since the nucleoplasmic lamin pool was found significantly reduced in the absence of LAP2α (*Dechat et al., 2000*; *Gesson et al., 2016*; *Gesson et al., 2014*; *Naetar et al., 2008*). However, the mechanisms remain unknown.

Here, we address the open question if and how LAP2α regulates the formation, properties and functions of nucleoplasmic A-type lamins. Surprisingly, we find that LAP2α is not involved in the generation of the nucleoplasmic lamin A/C pool from newly synthesized lamin A/C in interphase cells and from mitotic lamins in G1 phase, but instead regulates lamin A/C properties in a lamin A/C phosphorylation-independent manner. These findings led to a new perspective of LAP2α function in lamin A/C regulation, suggesting that LAP2α affects the state of lamin A/C assembly and/or their interactions in the nucleoplasm. We show that in the absence of LAP2α, lamins A and C in the nuclear interior are more resistant toward biochemical extraction, possibly through the formation of higher order structures and/or stronger interactions with nuclear components, leading to an

impaired detection by antibodies and decreased mobility. The impaired accessibility for antibodies against the lamin A/C N-terminus likely explains the previously observed reduction of nucleoplasmic lamins A/C in cells and tissues lacking LAP2α (*Naetar et al., 2008*). We propose that the interaction of soluble lamins with LAP2α in the nuclear interior maintains their mobile, low assembly state, which is important for proper lamin functions in nuclear organization and chromatin regulation.

## Results

### The nucleoplasmic lamin pool forms independently of LAP2α through at least two different processes

Previous observations using antibody labeling revealed a strong reduction of nucleoplasmic A-type lamins in cells lacking LAP2α, suggesting a prominent role for LAP2α in the formation or regulation of intranuclear lamins (*Naetar et al., 2008*). In order to elucidate the potential role of LAP2α in the formation of the nucleoplasmic pool of A-type lamins in interphase cells and during nuclear reassembly after mitosis, we performed live cell imaging of wildtype and LAP2α knockout HeLa cells (generated by CRISPR-Cas9, see *Figure 1—figure supplement 1*). First, we expressed GFP-pre-lamin A and imaged emerging pre-lamin A structures 5 hr post-transfection, when the newly expressed pre-lamin A undergoes post-translational processing (*Figure 1A*, columns 1 and 2, see also video files 1–3 associated with *Figure 1*). The nascent, transiently farnesylated pre-lamin A initially localizes to the nuclear periphery with very little visible nucleoplasmic staining (*Figures 1A*, 0-20'). Time-resolved quantification of the nucleoplasmic over peripheral fluorescence signal (*Figure 1A*, lower panel) reveals that lamins in the nuclear interior emerge gradually within 20 min, probably after their release from the nuclear periphery during further processing and removal of the C-terminal farnesyl group. Accordingly, an ectopically expressed, farnesylation-deficient mature lamin A (lacking its C-terminal 15 amino acids) is detectable prominently in the nucleoplasm at early timepoints when lamin structures first emerge (*Figure 1—figure supplement 2A*, see also video file 9 associated with *Figure 1—figure supplement 2*). Furthermore, an assembly-deficient lamin AΔK32 mutant (*Bank et al., 2011*; *Bertrand et al., 2012*) expressed in its pre-lamin form appears first at the nuclear periphery, before it translocates completely into the nucleoplasm (*Figure 1A*, third panel, see also video file 4 associated with *Figure 1*), while the farnesylation-deficient mature form of lamin AΔK32 mutant is exclusively in the nuclear interior throughout imaging (*Figure 1—figure supplement 2A*, see also video file 10 associated with *Figure 1—figure supplement 2*). These data suggest that post-translational processing rather than lamin filament assembly is the driving force for the initial localization of nascent pre-lamin A to the nuclear periphery.

In order to test whether LAP2α is involved in the subsequent release of the newly synthesized, processed lamin A from the lamina into the nuclear interior, we imaged nascent GFP-pre-lamin A in LAP2α knockout cells (*Figure 1A*). The increase in the lamin A signal in the nuclear interior occurred with similar kinetics in LAP2α knockout and wildtype cells, suggesting that fully processed lamin A/C can translocate to the nuclear interior independent of LAP2α.

To test whether LAP2α may affect the dynamic behavior of mature lamin A during the cell cycle, we analyzed GFP-pre-lamin A-expressing wildtype and LAP2α knockout HeLa cells 24 hr post transfection, when the majority of the ectopic pre-lamin A has been processed into the mature form (*Figure 1B*, see also video files 5–7 associated with *Figure 1*). As previously reported (*Broers et al., 1999*; *Moir et al., 2000*), lamin A structures completely disassemble at the onset of mitosis (*Figure 1B*, 0'), and, during nuclear reassembly, lamin A localizes uniformly throughout the nucleus in late telophase / early G1 phase (*Figure 1B*, 20'). Lamin A levels in the nuclear interior then gradually decrease within around 200 min due to relocation of the majority of lamins to the nuclear periphery, but a fraction of lamin A remains in the nuclear interior throughout interphase (*Figure 1B*, 40-240', see also *Dechat et al., 2000*; *Naetar et al., 2008*). As expected, ectopically expressed mature GFP-lamin A shows the same cell cycle-dependent behavior (*Figure 1—figure supplement 2B*, see also video file 11 associated with *Figure 1—figure supplement 2*). However, an assembly-deficient mutant, lamin AΔK32 remains fully nucleoplasmic throughout G1 phase, independent of whether the pre- or mature form was expressed (*Figure 1B* and *Figure 1—figure supplement 2B*, see also video files 8 and 12). Thus, lamin filament assembly is essential for the accumulation of A-type lamins at the nuclear periphery during G1 phase. In order to test the effect of LAP2α on the redistribution of

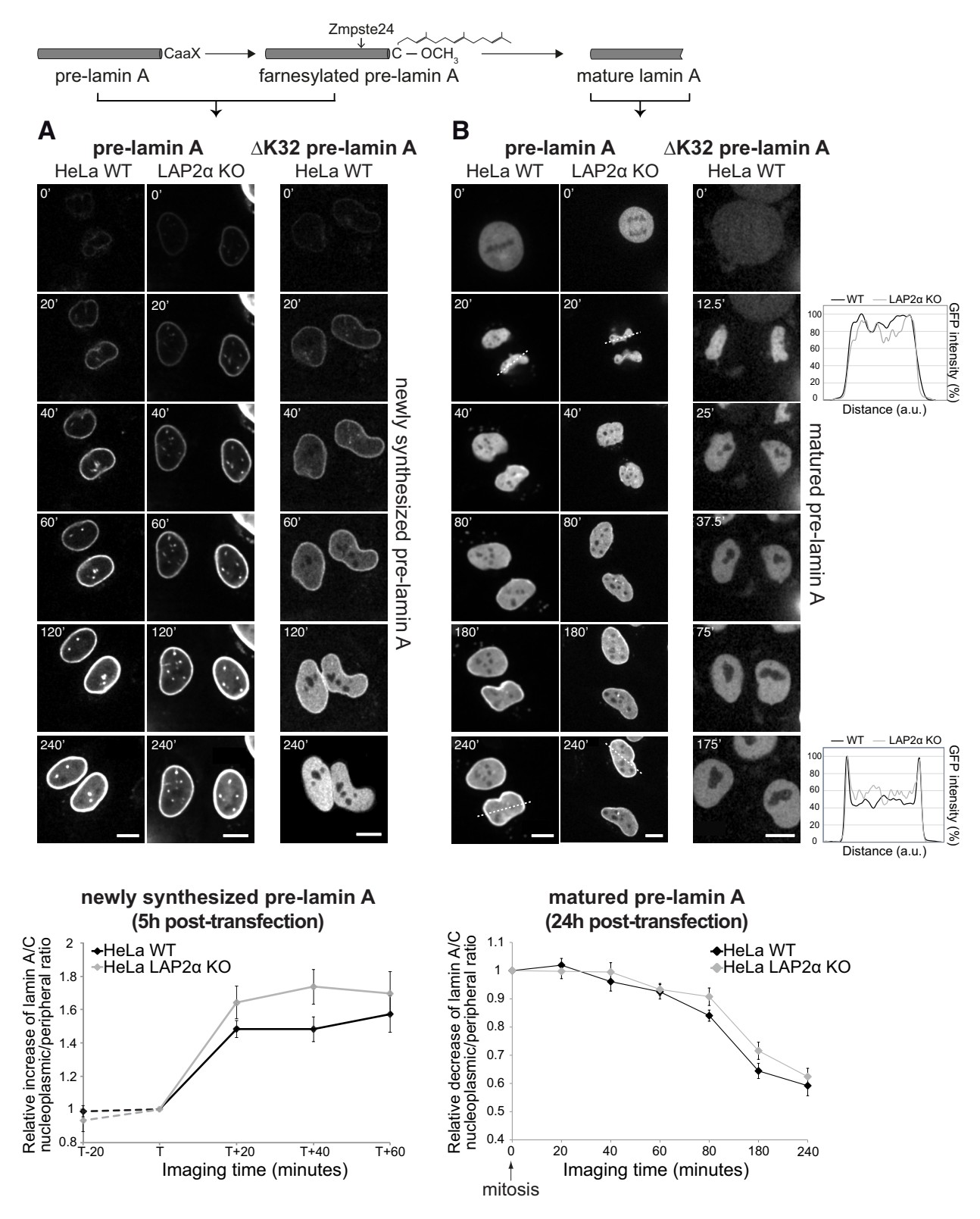

**Figure 1.** Absence of LAP2α does not affect formation of a nucleoplasmic pool of exogenously expressed GFP-lamin A. (**A**) Wildtype (WT) and LAP2α knockout (KO) HeLa cells (see *Figure 1—figure supplement 1*) were transiently transfected with EGFP-pre-lamin A or EGFP-ΔK32 pre-lamin A as indicated and analyzed by live-cell imaging 5 hr post-transfection. Schematic drawing on top explains lamin A processing state at the time of imaging. Time is indicated in minutes. See also video files: *Figure 1—videos 1* and *2*, corresponding to panels 1 and 2, *Figure 1—video 3* displaying an

*Figure 1 continued on next page*

*Figure 1 continued*

identically treated second HeLa LAP2α knockout clone, and *Figure 1—video 4*, corresponding to panel 3. Bottom: Ratio of nucleoplasmic to peripheral mean signal intensity of EGFP-lamin A was calculated (see Materials and methods for details) and normalized to time point 'T'. 'T' defines the moment preceding a significant increase of nucleoplasmic lamin A in each cell. 'T+/-x' indicates imaging time relative to 'T' with 'x' corresponding to minutes. Dashed lines indicate measurements performed in a smaller number of cells ($n_{wt}$ = 6, $n_{ko}$ = 6), demonstrating that nucleoplasmic lamin levels remain steadily low prior to the initial release of lamin A into the nucleoplasm (time point 'T'). Graphs display mean values ± S.E.M ($n_{wt}$ = 18, $n_{ko}$ = 23). p Values (repeated measures two-way ANOVA) are $p_{time}$ <0.0001 and $p_{genotype}$ = 0.2160 (non-significant). (**B**) WT and LAP2α KO HeLa cells were transfected as in (**A**), but imaged 24 hr post-transfection, where ectopically expressed lamin A is largely matured. See also video files: *Figure 1— videos 5* and *6*, corresponding to panels 1 and 2, *Figure 1—video 7* displaying an identically treated second HeLa LAP2α knockout clone, and *Figure 1—video 8*, corresponding to panel 3. Graphs on the right present examples for GFP-lamin A fluorescence intensity measured along the dashed lines in the images (20' and 240'). Bottom: Nucleoplasmic over peripheral signal ratio was determined using line profiles for all time points (see Materials and methods for details), normalized to mitosis/early G1 (time point 0') and plotted versus imaging time. Graphs display mean values ± S.E.M. ($n_{wt}$ = 10, $n_{ko}$ = 12). p Values (repeated measures two-way ANOVA) $p_{time}$ <0.0001 and $p_{genotype}$ = 0.2564 (non-significant). Scale bar: 10 μm.

The online version of this article includes the following video and figure supplement(s) for figure 1:

**Figure supplement 1.** Generation of LAP2α knockout HeLa cell clones using CRISPR-Cas9.

**Figure supplement 2.** Mature lamin A and mature ΔK32 lamin A initially localize to the nucleoplasm when newly expressed or when mitotically disassembled.

**Figure 1—video 1.** Wildtype HeLa cells were transiently transfected with EGFP-pre-lamin A and analyzed by live-cell imaging 5 hr post-transfection.
https://elifesciences.org/articles/63476#fig1video1

**Figure 1—video 2.** LAP2α knockout HeLa cells (clone sg1#2) were transiently transfected with EGFP-pre-lamin A and analyzed by live-cell imaging 5 hr post-transfection.
https://elifesciences.org/articles/63476#fig1video2

**Figure 1—video 3.** LAP2α knockout HeLa cells (clone sg2#25) were transiently transfected with EGFP-pre-lamin A and analyzed by live-cell imaging 5 hr post-transfection.
https://elifesciences.org/articles/63476#fig1video3

**Figure 1—video 4.** Wildtype HeLa cells were transiently transfected with EGFP-ΔK32 pre-lamin A and analyzed by live-cell imaging 5 hr post-transfection.
https://elifesciences.org/articles/63476#fig1video4

**Figure 1—video 5.** Wildtype HeLa cells were transiently transfected with EGFP-pre-lamin A and analyzed by live-cell imaging 24 hr post-transfection.
https://elifesciences.org/articles/63476#fig1video5

**Figure 1—video 6.** LAP2α knockout HeLa cells (clone sg1#2) were transiently transfected with EGFP-pre-lamin A and analyzed by live-cell imaging 24 hr post-transfection.
https://elifesciences.org/articles/63476#fig1video6

**Figure 1—video 7.** LAP2α knockout HeLa cells (clone sg2#25) were transiently transfected with EGFP-pre-lamin A and analyzed by live-cell imaging 24 hr post-transfection.
https://elifesciences.org/articles/63476#fig1video7

**Figure 1—video 8.** Wildtype HeLa cells were transiently transfected with EGFP-ΔK32 pre-lamin A and analyzed by live-cell imaging 24 hr post-transfection.
https://elifesciences.org/articles/63476#fig1video8

**Figure 1—video 9.** Wildtype HeLa cells were transiently transfected with EGFP-mature lamin A and analyzed by live-cell imaging 5 hr post-transfection.
https://elifesciences.org/articles/63476#fig1video9

**Figure 1—video 10.** Wildtype HeLa cells were transiently transfected with EGFP-ΔK32 mature lamin A and analyzed by live-cell imaging 5 hr post-transfection.
https://elifesciences.org/articles/63476#fig1video10

**Figure 1—video 11.** Wildtype HeLa cells were transiently transfected with EGFP-mature lamin A and analyzed by live-cell imaging 24 hr post-transfection.
https://elifesciences.org/articles/63476#fig1video11

**Figure 1—video 12.** Wildtype HeLa cells were transiently transfected with EGFP-ΔK32 mature lamin A and analyzed by live-cell imaging 24 hr post-transfection.
https://elifesciences.org/articles/63476#fig1video12

---

lamins from the nuclear interior to the periphery after mitosis, we performed live cell imaging of LAP2α knockout cells. Quantification of the nucleoplasmic over peripheral GFP-lamin A signal after mitosis revealed a similar gradual decrease in nucleoplasmic staining during G1 phase in LAP2α knockout and wildtype cells, and more surprisingly, the steady state levels of lamin A in the nuclear interior throughout interphase were unaffected in LAP2α knockout versus wildtype cells (*Figure 1B*, lower panel).

In conclusion, we show that nucleoplasmic A-type lamins are generated by two different pathways: (i) newly synthesized pre-lamin A redistributes from the periphery to the nuclear interior upon post-translational processing and (ii) a fraction of soluble A-type lamins of the previous mitosis remains in the nucleoplasm during lamina assembly in G1 phase. However, none of these processes were significantly altered in the absence of LAP2α, suggesting that nucleoplasmic A-type lamins form independently of LAP2α. In addition, steady-state lamin A levels in the nuclear interior in interphase apparently were unchanged in the absence of LAP2α. These findings are conflicting with our previous results showing loss of nucleoplasmic lamin A/C in LAP2α knockout cells in immunofluorescence microscopy (*Naetar et al., 2008*).

## LAP2α-deficiency does neither affect the formation nor the steady state level of endogenous lamin A in the nuclear interior

We reasoned that the lack of any effect of LAP2α loss on the formation of the nucleoplasmic lamin A pool may be due to overexpression of the ectopic GFP-lamin A. Thus, in order to create a more physiological setting, we tagged the endogenous lamin A gene with mEos3.2, a monomeric variant of the photoconverter EosFP (*Zhang et al., 2012*), in wildtype and LAP2α knockout mouse fibroblasts using CRISPR-Cas9 (*Figure 2A*, *Figure 2—figure supplements 1* and *2*). Following a rigid testing of the wildtype clones (see details in *Figure 2—figure supplement 1*), we picked a clone (WT clone #21) for further analyses, which is tetraploid and expresses tagged lamin A/C from 1 allele and untagged lamin A/C from three alleles in a 1:3 ratio at the transcript level (*Figure 2B,C* and *Figure 2—figure supplement 1F,G*). At the protein level, this ratio shifts to 1:8, suggesting that the addition of the mEos3.2-tag leads to a slight destabilization of lamin A and C protein (*Figure 2—figure supplement 1G*). However, the tagged lamins behave like untagged proteins (see below) and the level of tagged lamin A/C proteins was sufficient for high-quality microscopic analysis (*Figure 2C–H*). We then generated isogenic LAP2α knockout clones from this WT clone by CRISPR-Cas9 (for details see *Figure 2—figure supplement 2*). Both WT and LAP2α knockout clones display normal nuclear morphology and the mEos3.2-tagged lamin A/C correctly localizes at the nuclear periphery and in the nuclear interior (*Figure 2C*) and was stably integrated into the nuclear lamina, as demonstrated by photoconversion of tagged lamins, where no significant mobility of photo-converted lamins in the lamina was observed for several hours (*Figure 2D*). Moreover, the solubility in high-salt and detergent-containing buffers was comparable between untagged and tagged lamins A and C (*Figure 2—figure supplement 1G*). Thus, endogenously tagged lamin A/C behaves like untagged protein in terms of localization, dynamics and biochemical properties (*Figure 2* and *Figure 2—figure supplement 1*), supporting its functionality. Notably, cells with all *Lmna* alleles tagged, thus expressing only mEos3-2-lamin A/C, displayed similar viability, nuclear morphology, and lamina structure when compared to parental control cells expressing only WT lamin A/C (*Figure 2—figure supplement 1H and I*).

To investigate the potential role of LAP2α in the retention of endogenous lamins A and C in the nucleoplasm in G1 phase, and in controlling steady-state levels of lamin A/C in the nuclear interior throughout interphase, we followed mEos3.2-tagged lamin A/C in wildtype and LAP2α knockout clones by live cell imaging. We performed time lapse studies starting at the exit from mitosis through G1 phase, and determined the nucleoplasmic to peripheral lamin A/C signal ratio in hundreds of cells by automated quantification using a custom-made FIJI plugin (*Figure 2E*). Neither the post-mitotic dynamics of nucleoplasmic lamin re-localization nor its steady-state levels in late G1 phase were affected in LAP2α knockout versus wildtype cells (*Figure 2F*).

To further analyze nucleoplasmic lamin A/C levels through interphase, we fixed mEos3.2-lamin A/C expressing wildtype and LAP2α knockout cells and automatically quantified their nucleoplasmic (N) to peripheral (P) lamin A/C ratio as described before. The lamin A/C N/P ratio was then plotted against the intensity of the DAPI signal to stage cells within the cell cycle (*Roukos et al., 2015*; *Figure 2G*). As expected, the nucleoplasmic to peripheral lamin A/C ratio was higher in G1 phase (WT G1:0.26 ± 0.0055) than in S/G2 phase (WT S/G2: 0.19 ± 0.0045, Dunn's post test p<0.0001), but both ratios were unchanged in the absence of LAP2α (LAP2α KO G1: 0.29 ± 0.0054, S/G2: 0.21 ± 0.0044). The observed subtle variability is likely due to clonal differences, as it was also observed between different wildtype clones (e.g. LAP2α WT sg1 ctrl G1: 0.27 ± 0.0045, S/G2: 0.20 ± 0.0036) (*Figure 2G and H*). To ensure specificity of the nucleoplasmic signal and investigate the influence of out-of-focus light from the much stronger peripheral lamin A/C fluorescence, we

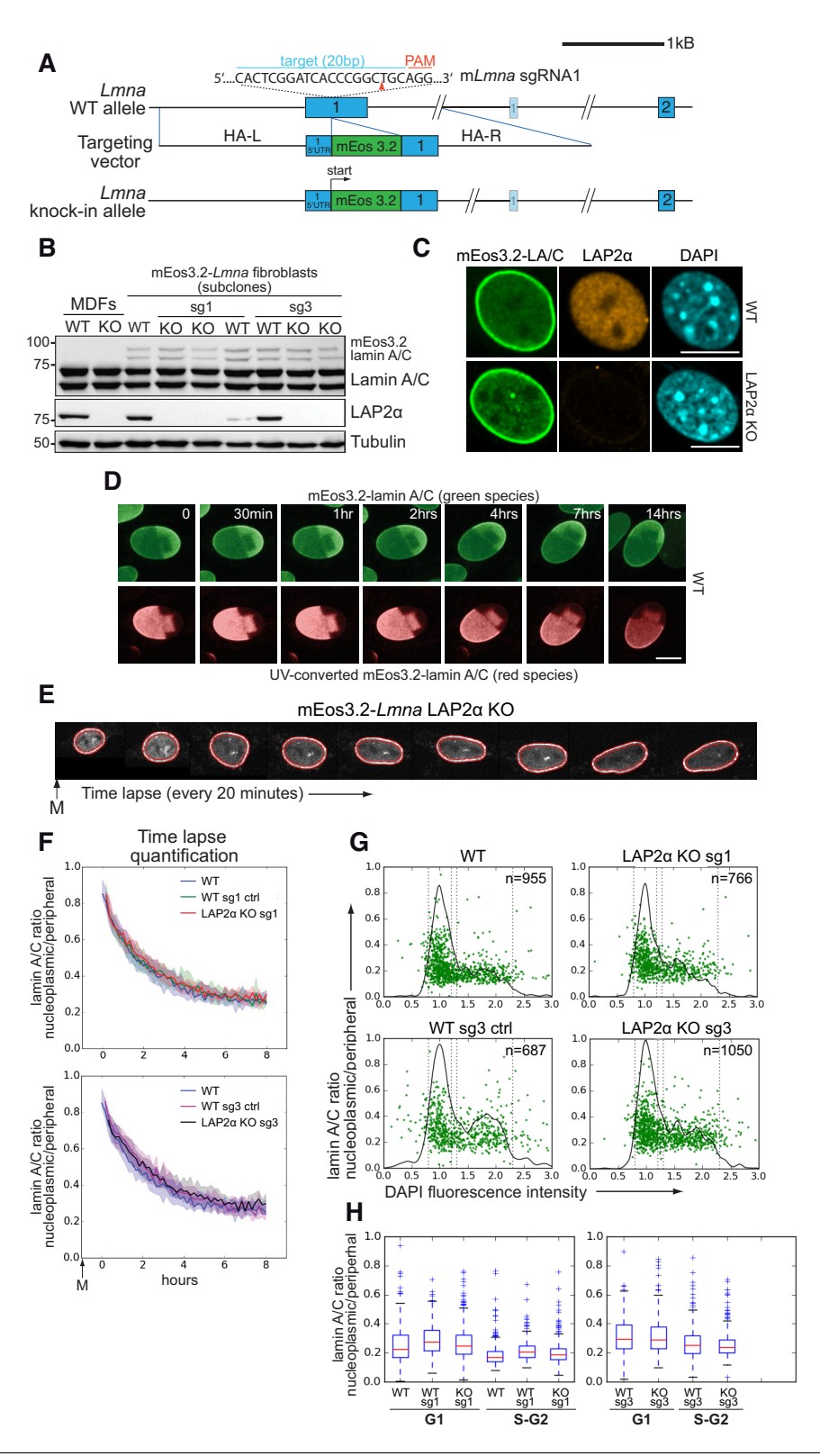

**Figure 2.** Absence of LAP2α does not alter endogenous nucleoplasmic lamin A/C levels. (**A**) Schematic view of exons 1 and 2 (bars) and adjacent introns of the mouse *Lmna* locus (top), the targeting construct provided as homology-directed repair template after Cas9-mediated double strand break (middle), and the knock-in allele after successful integration of mEos3.2 (bottom). Very long introns are not displayed in their original length as indicated by a double slash. The second, light-colored exon 1 encodes the N-terminus of meiosis-specific lamin C2. The target sequence of *Lmna-*

*Figure 2 continued*

specific sgRNA1 in exon 1 is shown (blue). Protospacer-adjacent motif (PAM) is marked in red. Red arrowhead: expected Cas9 cut site. HA-L: Homology Arm-Left; HA-R: Homology Arm-Right. For further cell characterization see *Figure 2—figure supplement 1*. (B) mEos3.2-*Lmna* clone derived from wildtype (WT) mouse dermal fibroblasts (MDFs) and subclones generated after treatment with LAP2α-specific sgRNAs 1 or 3 (see *Figure 2—figure supplement 2*) were processed for western blotting using antibodies to the indicated antigens (anti lamin A/C E1, anti LAP2α 1H11). Positions of mEos-tagged and untagged lamins A and C are indicated. Wildtype and LAP2α knockout (KO) MDFs were added as controls. (C) mEos3.2-*Lmna* WT and LAP2α KO cells were processed for immunofluorescence microscopy using antibodies against LAP2α (1H11) and DAPI to visualize DNA. Confocal images are shown. Bar: 10 μm. (D) mEos3.2-*Lmna* WT cells were exposed to UV light (405 nm) to convert mEos3.2-lamin A/C from green to red. Cells were then tracked for 14 hr by live-cell imaging to test for stable integration of mEos3.2-lamin A/C into the peripheral lamina. No significant mobility of converted lamins in the lamina was observed over the entire imaging time. Pictures display merged Z-stacks to visualize most of the peripheral lamina. Bar: 10 μm. (E) Representative example of a LAP2α KO cell expressing mEos3.2-lamin A/C tracked by live-cell imaging throughout G1, followed by automated calculation of nucleoplasmic over peripheral lamin A/C ratio ($t_0$: first available image after mitosis). The area corresponding to the nuclear lamina is delineated by red lines and was defined by a custom-made algorithm implemented in FIJI/ImageJ software. Average values for isogenic mEos3.2-*Lmna* LAP2α WT and KO clones were plotted in curves (F). The designation 'sg1' and 'sg3' refers to the sgRNA used to generate that clone (see *Figure 2—figure supplement 2*). Clones treated with sgRNAs, but still expressing LAP2α, were included as additional controls and are designated as 'WT sg ctrl'. The 20th and 80th percentile are displayed for each curve (shaded area). $n_{WT}$ = 129, $n_{WTsg1ctrl}$=159, $n_{KOsg1}$ = 106, $n_{WTsg3ctrl}$=111, $n_{KOsg3}$ = 147 (G) Isogenic clones were fixed and DNA was stained with DAPI. Nucleoplasmic (N) to peripheral (P) signal intensity of mEos3.2-Lamin A/C was calculated for single cells (green dots) using a custom-made FIJI plugin and plotted versus DAPI fluorescence intensity, indicative of cell cycle stage (see also *Figure 2—source data 1*). Black line outlines number of cells versus DAPI intensity (histogram). Vertical dotted lines indicate what was classified as G1 and S-G2 in (H). (H) Single-cell N/P ratios from (G) were averaged for G1 and S/G2 and are shown in a box plot (median in red within the first and third quartiles; whiskers: minimal and maximal datapoint excluding outliers). The specificity of nucleoplasmic lamin A/C signal was demonstrated by comparison to lamin B1 N/P ratios (see *Figure 2—figure supplement 3*).

The online version of this article includes the following source data and figure supplement(s) for figure 2:

**Source data 1.** Nucleoplasmic to peripheral signal intensity ratio and DAPI fluorescence intensity of mEos3.2-Lamin A/C WT and LAP2α KO cells.
**Figure supplement 1.** Characterization of mEos3.2-*Lmna* mouse fibroblasts.
**Figure supplement 2.** Generation of isogenic LAP2α knockout mEos3.2-*Lmna* clones using CRISPR-Cas9.
**Figure supplement 3.** Assessing specificity of nucleoplasmic lamin A/C fluorescence signal by quantifying fluorescent out-of-focus bleed-through by the nuclear lamina.

---

determined N/P ratios also for lamin B1, which is exclusively localized at the periphery (*Figure 2—figure supplement 3*). When lamin B1 ratios, which would represent out-of-focus signal in the nucleoplasm, were substracted from lamin A/C N/P ratios, the resulting values were still significantly above 0 (*Figure 2—figure supplement 3*, lower panel), indicating that the detected nucleoplasmic lamin A/C signal stems, at least to a large extent, from lamins in the nuclear interior and not from out-of-focus light from the nuclear lamina.

Thus, endogenous lamins A and C are retained in the nucleoplasm after mitosis and remain in the nuclear interior throughout interphase independent of LAP2α, which is in stark contrast to previous findings demonstrating a clear reduction of nucleoplasmic lamin A/C signal in LAP2α knockout cells and tissues by immunofluorescence microscopy (*Naetar et al., 2008*).

## LAP2α loss changes properties of nucleoplasmic lamin A/C rendering them less mobile and more resistant to biochemical extraction

In order to resolve these contradicting results, we first tested mEos3.2-lamin A/C-expressing wild-type cells and isogenic LAP2α knockout clones by immunofluorescence microscopy using an antibody to the lamin A/C N-terminus (N18), which, unlike an antibody to the lamin A/C C-terminus (3A6, see *Figure 3—figure supplement 1B*), preferentially stains nucleoplasmic lamins A and C (*Gesson et al., 2016*; *Figure 3A and B*, *Figure 3—figure supplement 1A*). While the intranuclear mEos3.2 fluorescence signal was similar in LAP2α knockout and wildtype cells, the nucleoplasmic lamin A/C antibody-derived signal was clearly reduced in LAP2α knockout versus wildtype cells (*Figure 3A,B* and *Figure 3—figure supplement 1A*, compare red and green channels and see quantification in *Figure 3D*). Interestingly, the intranuclear antibody staining in LAP2α knockout cells completely recovered following post-fixation treatment with DNase and RNase (*Figure 3C and D*), suggesting epitope masking specifically in LAP2α knockout cells.

In order to exclude the possibility that mEos3.2-lamin A/C protein behaves differently as compared to untagged lamin A/C (both are expressed in the WT cell clone), we tested other cell clones expressing exclusively mEos3.2-tagged lamin A/C (without any untagged lamin A/C, *Figure 3—*

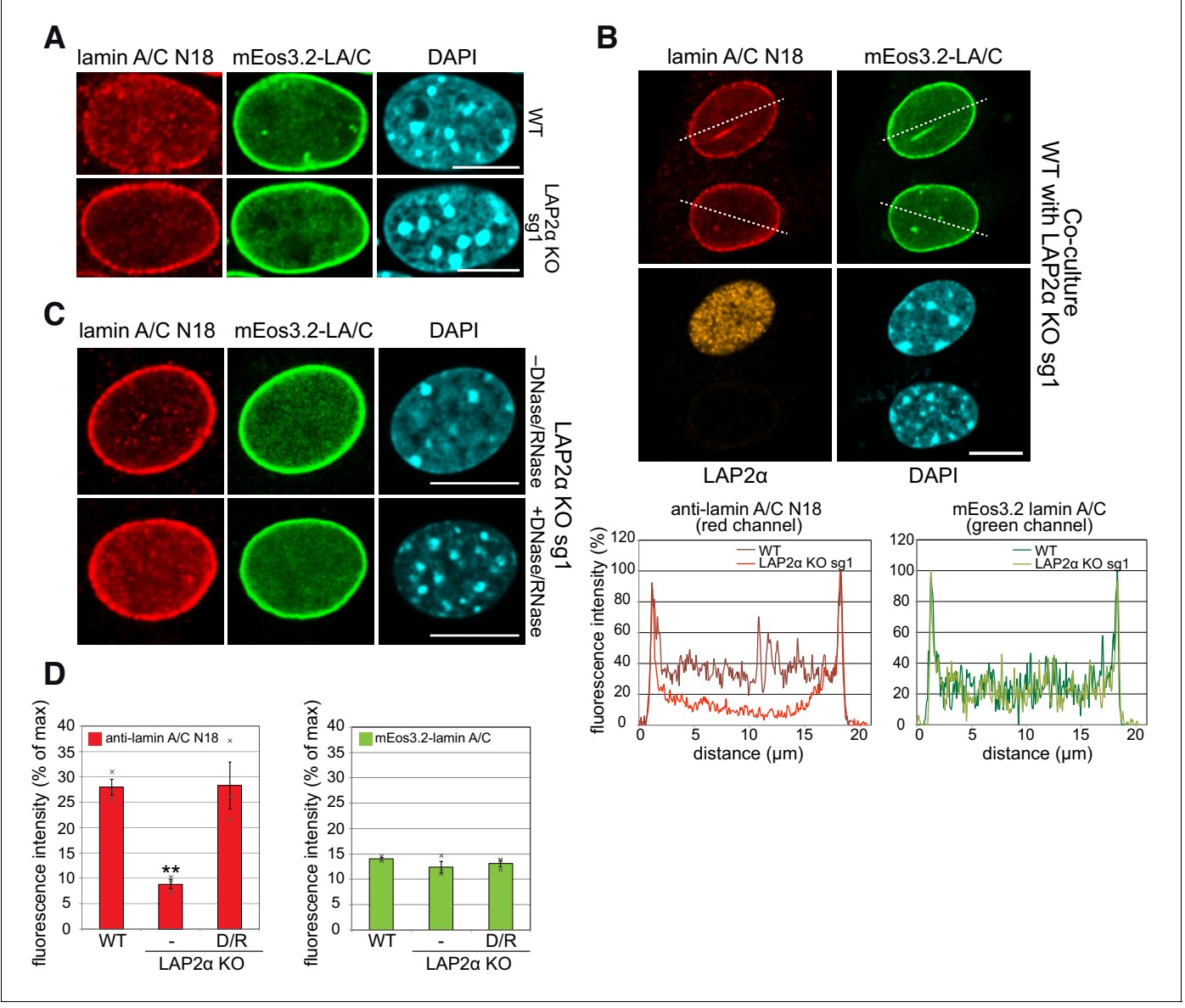

**Figure 3.** Absence of LAP2α reduces nucleoplasmic lamin A/C staining in immunofluorescence. (**A**) mEos3.2-*Lmna* WT and LAP2α KO sg1 cells were processed for immunofluorescence microscopy using antibody N18 against lamin A/C N-terminus, and DAPI to visualize DNA. Lamin A/C antibody N18 preferably stains nucleoplasmic lamin A/C. (**B**) mEos3.2-*Lmna* WT and LAP2α KO sg1 cells were co-cultured and processed for immunofluorescence microscopy as in (**A**) using antibodies N18 against lamin A/C and 1H11 against LAP2α, and DAPI to visualize DNA. Fluorescence intensity was determined for each cell along the dashed line in the red and green channel and is depicted in percent of maximum value in the graphs below. (**C**) LAP2α KO sg1 cells were processed as in (**A**) using lamin A/C antibody N18 without or with prior treatment with DNase I and RNase A to reverse N18 epitope masking. Bar: 10 µm. (**D**) Average nucleoplasmic fluorescence intensity from 3 wildtype and 3 LAP2α KO clones with and without prior DNase/RNase (D/R) treatment for N18 stainings and for mEos3.2 lamin A/C signal is depicted. Data represent averages ± S.E.M. Single data points are shown for each group. One-way ANOVA p value for N18 is 0.0020, F = 20.85; **p<0.01 (Tukey's post-hoc test: WT vs. KO: p=0.0034; KO vs KO plus DNase/RNase: p=0.0033); p value for mEos3.2-lamin A/C is 0.3694 (non-significant), F = 1.181.
The online version of this article includes the following figure supplement(s) for figure 3:

**Figure supplement 1.** Nucleoplasmic lamin A/C antibody staining but not the mEos3.2 fluorescence signal is reduced in LAP2α knockout versus wild-type fibroblasts in fluorescence microscopy.

*figure supplement 1C and D*). These clones exhibit the same properties and fitness as parental cells expressing only untagged WT lamin A/C (see *Figure 2—figure supplement 1H and I*) and display a similar reduction in N18 antibody-derived nucleoplasmic lamin staining in the absence of LAP2α, while the mEos3.2-lamin A/C fluorescence signal in the nuclear interior was unchanged (*Figure 3— figure supplement 1C*). Moreover, peripheral lamin A/C staining using the 3A6 antibody was unchanged in the absence of LAP2α in both, cells expressing only mEos3.2-lamin A/C and parental cells (*Figure 3—figure supplement 1D*).

Overall, these data suggest that – instead of a complete absence of nucleoplasmic A-type lamins – the reduced staining in LAP2α knockout cells reflects an alteration of nucleoplasmic lamin A/C properties that led to masking of the N-terminal epitope recognized by the antibody.

To test potential changes in nucleoplasmic lamin A/C properties in LAP2α knockout versus wild-type cells, we first tested lamin mobility. While lamins at the periphery form stable filaments, nucleo-plasmic lamin A/C was shown to be highly mobile (*Broers et al., 1999*; *Kolb et al., 2011*; *Shimi et al., 2008*). We tested the mobility of nucleoplasmic lamins by Continuous Photobleaching (CP), where the fluorescence intensity is measured in a small spot within the nucleoplasm of mEos3.2-lamin A/C expressing cells at a high frequency over time. This analysis revealed two fluores-cent sub-populations with different mobility: an immobile fast bleaching fraction, and a highly mobile slow bleaching fraction (*Figure 4A*, left panel). As previously demonstrated (*Bronshtein et al., 2015*), 10–40% of lamin A in the nucleoplasm is immobile in the majority of wildtype cells (*Figure 4A*). Intriguingly, in the absence of LAP2α we observed a noticeable shift toward a higher fraction of immobile lamin A/C in the nuclear interior in the range of 30 up to 60% (*Figure 4A*, right panel). Thus, the mobile fraction of lamin A in the nuclear interior is reduced in LAP2α knockout ver-sus wildtype cells.

In addition, we measured the mobile lamin A/C fraction in the nuclear interior by fluorescence correlation spectroscopy (FCS), which revealed a slower and a faster-moving fraction as previously reported (*Shimi et al., 2008*). Interestingly, although continuous photobleaching has clearly revealed an overall decrease in the relative amount of the mobile (moving) lamin A/C pool in LAP2α knockout cells, the small remaining mobile fractions showed increased diffusion (*Shimi et al., 2008*). The ratio of the FCS diffusion rates ($D_{WT}/D_{KO}$, 0.76 for the fast fraction) equals the cube root of the inverted molecular weight ratio ($M_{KO}/M_{WT}$) of the nucleoplasmic lamin complex (see Materials and methods for details). Assuming that the WT complex contains a 1:1 ratio (our unpublished data) of mEos3.2-lamin A and LAP2α (100 kDa + 75 kDa), whereas LAP2α KO cells contain only mEos3.2-lamin A (100 kDa), the calculated cube root of the KO/WT molecular weight ratio is 0.83. Therefore, the measured FCS diffusions may be explained by the loss of LAP2α in the lamin A/C complex in LAP2α knockout cells (*Figure 4B*).

To reveal further changes in lamin A properties in LAP2α knockout versus wildtype cells, we per-formed biochemical extraction in buffers containing detergent and 150–500 mM salt. Lamins A and C were significantly less extractable in the absence of LAP2α in both, mouse fibroblasts and HeLa cells (*Figure 4C* and *Figure 4—figure supplement 1A*), consistent with a shift of nucleoplas-mic lamins into more stable, possibly higher assembly structures or with stronger interactions of lamin complexes with nuclear structures upon loss of LAP2α. Next, we aimed at visualizing poten-tially higher assembly structures in LAP2α knockout cells by high-resolution microscopy. While microscopy of untreated cells did not reveal clear differences in the appearance of nucleoplasmic lamin structures (*Figure 4D*, upper panel), extraction of cells in detergent and salt-containing buffers prior to fixation revealed significantly more and larger, extraction-resistant lamin A structures in the nuclear interior in LAP2α knockout versus wildtype mEos3.2-lamin A/C cells (*Figure 4D* and quantifi-cation in *Figure 4E*). Strikingly, while in wildtype cells these structures appeared as few punctae, LAP2α knockout cells displayed significantly more stable structures of an extended, possibly filamen-tous appearance (*Figure 4D and E*).

Altogether, loss of LAP2α does not lead to a reduction of the nucleoplasmic pool of lamin A/C, it rather affects its mobility, interaction and/or assembly state.

## LAP2α binds to intranuclear lamin A/C and inhibits its assembly

In order to better understand how LAP2α might influence the mobility and/or assembly state of nucleoplasmic lamin A/C, we performed in vitro interaction analyses of these proteins. Bacterially expressed, purified recombinant lamin A and LAP2α in urea-containing buffer, were dialyzed alone

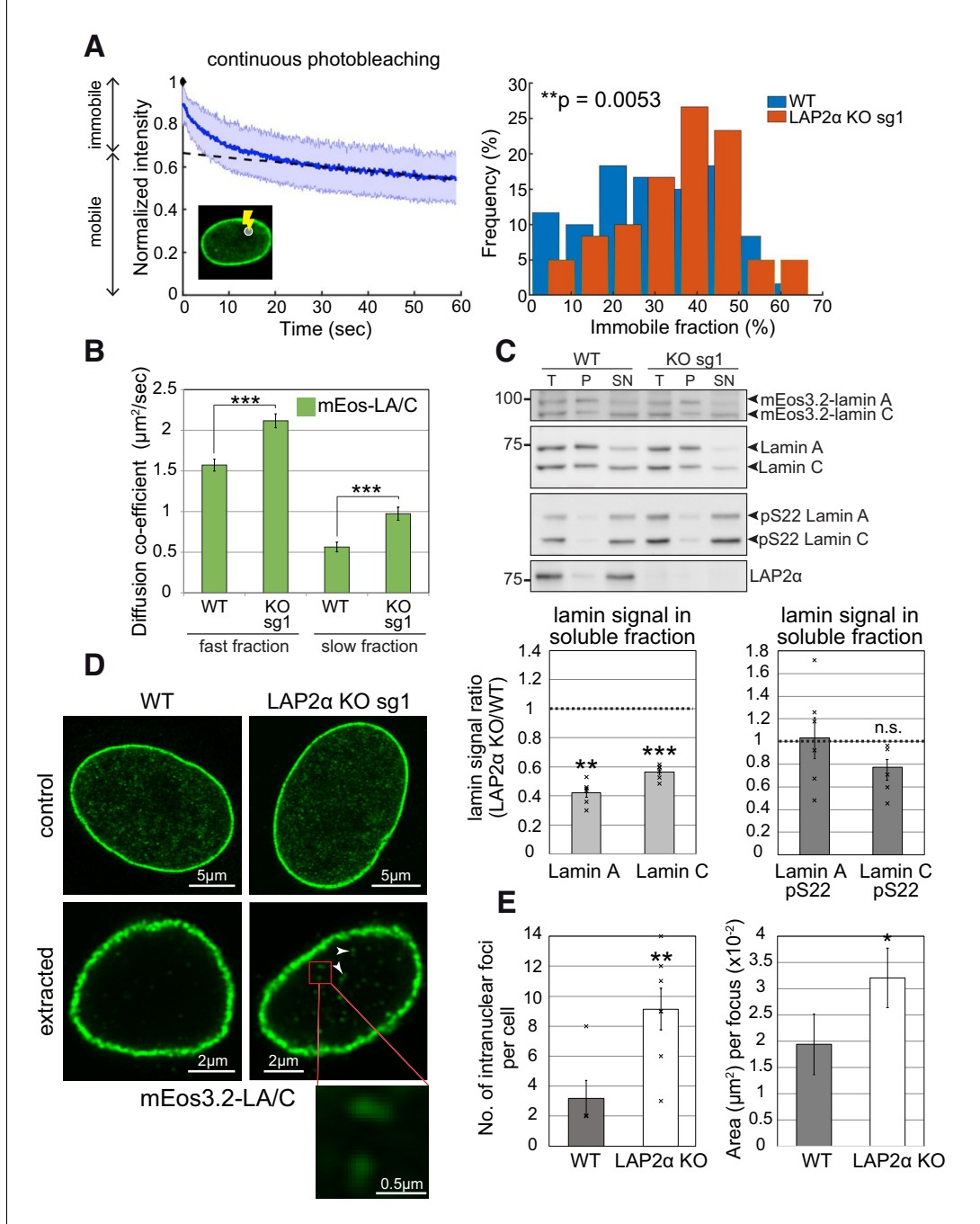

**Figure 4.** Nucleoplasmic lamin A/C is less mobile, more extraction-resistant and form larger assemblies in the absence of LAP2α. (**A**) Left: Depicted is a representative curve obtained by continuous photobleaching (CP, see schematic inset on the lower left) using mEos3.2-lamin A/C WT cells, allowing to determine mobile and immobile fractions (shown on the left side of the graph) of the measured mEos3.2-lamin A/C. Blue line represents normalized average intensity of mEos3.2-lamin A/C measured over time from a selected spot inside the nucleus. Right: Immobile fractions of mEos3.2-lamin A and C protein as calculated from measured CP curves of mEos3.2-lamin A/C WT and LAP2α KO sg1 cells are depicted as histogram (see also **Figure 4— source data 1**). $n_{WT}$ = 59, $n_{KO}$ = 60, p=0.0053 (two-tailed student's t test of arcsin transformed values). (**B**) Graph displays the diffusion co-efficient of fast and slow moving mEos3.2-lamins A and C as determined by fluorescence correlation spectroscopy (FCS) measurements (see also **Figure 4— source data 2**). Data represent averages ± S.E.M.; $n_{WT (fast fraction)}$=106, $n_{WT (slow fraction)}$=96, $n_{KO (fast fraction)}$=97, $n_{KO (slow fraction)}$=99; ***p<0.0001 (Mann-Whitney U test). (**C**) Cells from isogenic WT and LAP2α KO clones were extracted in salt and detergent-containing buffer (500 mM NaCl, 0.5% NP-40). Extracts were processed for Western blot analysis. Upper panel shows representative western blots from WT and LAP2α KO sg1 cells using antibodies against total lamin A/C (E1), lamin A/C phosphorylated at serine 22 (pS22) or LAP2α (1H11). T: total lysate; P: insoluble pellet fraction; SN: soluble, extracted supernatant fraction. Western blots were quantified and lamin signal in the supernatant was normalized to total lamin A/C. Graphs display
*Figure 4 continued on next page*

*Figure 4 continued*

lamins A and C levels in the supernatant fraction of LAP2α knockout samples as average fold difference ± S.E.M over wildtype samples. Single data points are shown for each group. $n_{WT}$ = 6, $n_{KO}$ = 6; **p=0.00013, ***p=2.09E-5, n.s.: non-significant (p=0.072) (paired student's t-test on log transformed values). (D) mEos3.2-lamin A/C WT and LAP2α KO sg1 cells were processed for immunofluorescence microscopy with and without prior extraction in salt and detergent-containing buffer. Confocal super-resolution (Airyscan) images of mEos3.2-lamin A/C signal are depicted. Larger lamin A/C nucleoplasmic structures in LAP2α knockout cells are marked by arrowheads and displayed as larger inset (bottom). (E) Graphs show quantification of number (No.) of intranuclear lamin A/C structures per cell and mean area per structure. Data represent averages ± S.E.M. Left graph: $n_{WT}$ = 5, $n_{KO}$ = 7 (single data points are displayed), p=0.0076 (Mann-Whitney U test); right graph: $n_{WT}$ = 16, $n_{KO}$ = 64, p=0.0433 (Mann-Whitney U test); *p<0.05, **p<0.01.

The online version of this article includes the following source data and figure supplement(s) for figure 4:

**Source data 1.** Immobile mEos3.2-lamin A/C fraction in the nucleoplasm of WT and LAP2α KO cells determined by constant photobleaching.

**Source data 2.** Diffusion co-efficient of fast and slow moving mEos3.2-lamins A and C in WT and LAP2α KO cells as determined by FCS.

**Figure supplement 1.** Nucleoplasmic lamins A and C are more resistant to extraction in the absence of LAP2α in HeLa cells without detectable changes in lamin phosphorylation.

or together into lamin assembly buffer containing 300 mM NaCl, a salt concentration where lamin A does not form higher assembly structures (*Foeger et al., 2006*), and analyzed the protein samples by sucrose gradient centrifugation. (*Figure 5A*). While both proteins alone sedimented mostly to fractions 2 and 3 of the sucrose gradient, their sedimentation pattern shifted toward lower fractions in the protein mix, consistent with the formation of larger hetero-complexes (*Figure 5A*). As a control, a LAP2α mutant lacking the C-terminus that mediates interaction with A-type lamins (*Dechat et al., 2000*) did not alter the sedimentation pattern of lamin A (*Figure 5B*). Next, the proteins were dialyzed into buffer with a lower salt concentration (100 mM NaCl) allowing formation of higher assembly lamin structures (*Foeger et al., 2006*). High-molecular-mass lamin A assemblies were separated from those of lower oligomeric states by centrifugation and analyzed by gel electrophoresis (*Figure 5C*). Strikingly, the presence of LAP2α reduced the amount of pelleted high-molecular-mass lamin A assemblies by half, indicating that LAP2α impairs lamin A assembly into larger structures (*Figure 5C*). Thus, binding of LAP2α to lamin A may directly impair the formation of higher order lamin assemblies.

To confirm that LAP2α and lamin A/C interact in vivo in mEos3.2-lamin A/C cell lines, we performed co-immunoprecipitation using different lamin A/C antibodies. In accordance with previous data (*Dechat et al., 2000*; *Gesson et al., 2016*), antibodies to the lamin A/C N-terminus co-precipitated LAP2α from cell extracts (*Figure 5D*, antibodies N18 and E1), whereas an antibody to the lamin A/C C-terminus failed to co-precipitate LAP2α (*Figure 5D*, antibody 3A6), likely because binding of LAP2α to the lamin A/C C-terminus masks the epitope recognized by this antibody (Gesson et a., 2016). Thus, LAP2α interacts with lamin A/C in vivo and may thereby reduce the formation of higher order lamin assemblies in the nuclear interior.

Phosphorylation is another process potentially regulating lamin assembly and mobility, as particularly lamin phosphorylation at serines at position 22 and 392 was described to regulate lamina disassembly during mitosis, as well as lamin A/C localization, mobility and solubility in interphase cells (*Heald and McKeon, 1990*; *Kochin et al., 2014*). Thus, we wanted to test whether LAP2α may affect lamin A/C assembly indirectly, by regulating its phosphorylation status. Antibodies specifically recognizing lamin A/C phosphorylated at S22 revealed a similar punctate nucleoplasmic staining with complete absence of peripheral lamina staining in both LAP2α wildtype and knockout cells (*Figure 5E* and *Figure 4—figure supplement 1C*; *Kochin et al., 2014*) and total levels of pS22 lamin A/C and pS392 lamin A/C were unchanged in knockout versus wildtype cells (*Figure 5E and F*, *Figure 4—figure supplement 1B and C*). In addition, the relative level of detergent/salt-extractable pS22 lamins A and C were unchanged in LAP2α knockout versus wildtype cells, in stark contrast to the significantly reduced levels of extractable total lamin A/C (*Figure 4C*). Thus, LAP2α does neither affect the levels nor properties of phosphorylated lamins A and C. However, LAP2α can bind to pS22 lamin A/C as shown by coprecipitation from cell lysates (*Figure 5D*, antibody pS22). We concluded that LAP2α binds both un(der)phosphorylated and pS22 lamin A/C, but affects preferentially the assembly state of un(der)phosphorylated lamin A/C, keeping it in a mobile, lower assembly state in a phosphorylation-independent manner.

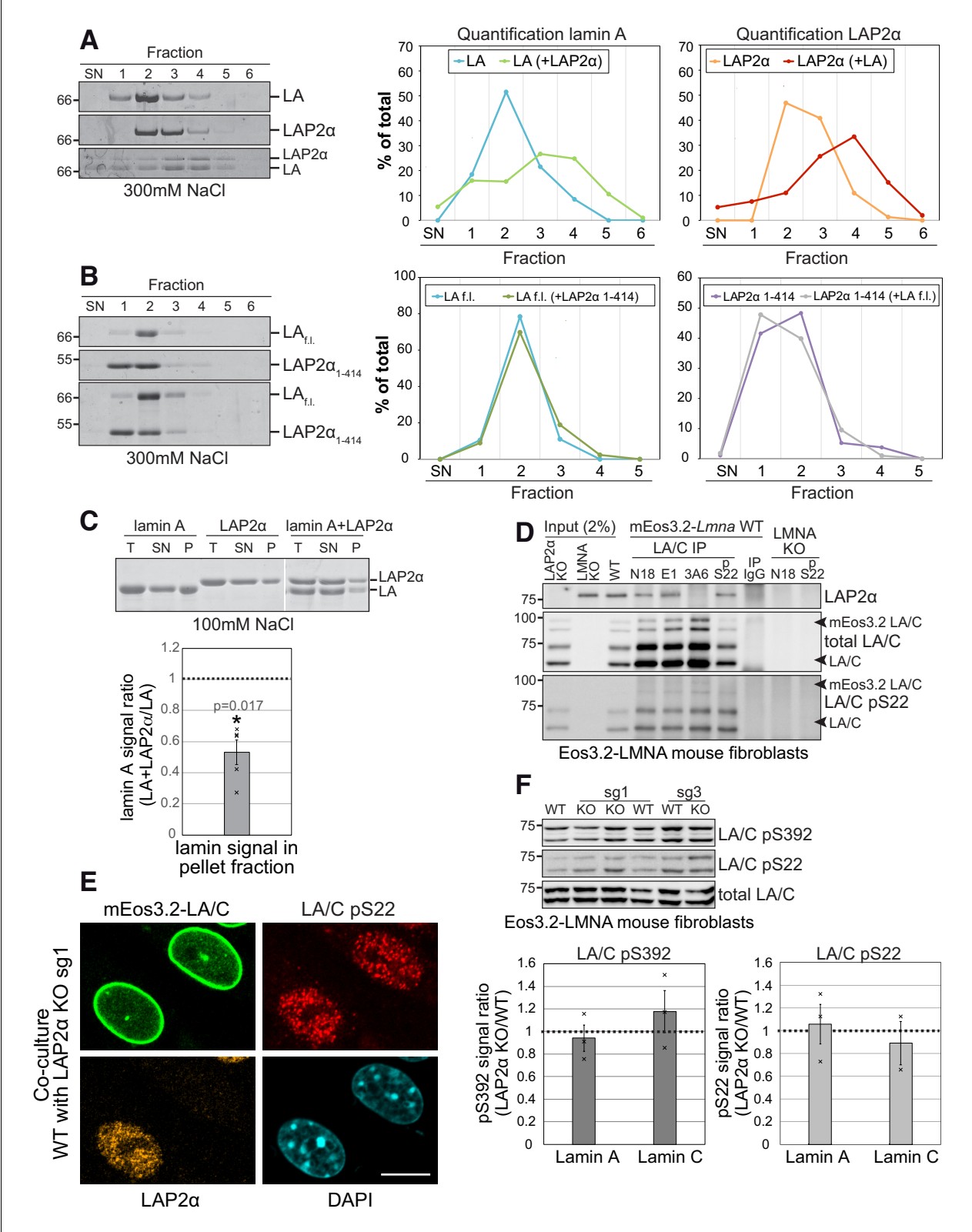

**Figure 5.** LAP2α binds to lamin A/C and inhibits their assembly without altering lamin phosphorylation. (**A, B**). Purified recombinant lamin A and LAP2α full length (f.l.) (**A**) or lamin A-binding mutant LAP2α$_{1-414}$ (**B**) were dialyzed either alone or together into assembly buffer with 300 mM NaCl. Samples were separated on a 10% to 30% sucrose gradient, followed by collection of fractions and quantification of protein bands. Exemplary Coomassie stained gels of fractions ‚supernatant' (SN) to six are shown on the left. The calculated protein amount per fraction (% of total protein) was plotted as

*Figure 5 continued on next page*

*Figure 5 continued*

curve chart on the right. (C) Lamin A and LAP2α were dialyzed either alone or together into assembly buffer as in (D), but with 100 mM NaCl, enabling formation of higher assembly lamin structures. After centrifugation, total (T), supernatant (SN) and pellet (P) fractions were analyzed on a Coomassie gel, protein bands were quantified and normalized to total protein levels. Graph displays lamin A levels in the pellet fraction of mixed samples (lamin A + LAP2α) as average fold difference ± S.E.M over samples with lamin A alone, n = 5 (single data points are displayed). (D) Lamin A/C was immunoprecipitated from mEos3.2-lamin A/C WT cells and LAP2α and lamin A/C knockout controls using the indicated antibodies recognizing different regions of lamin A/C. Immunoprecipitates were analyzed by western blotting using the indicated antibodies (anti lamin A/C 3A6, anti pS22 lamin A/C, anti LAP2α 1H11). (E) mEos3.2-*Lmna* WT and LAP2α KO sg1 cells were co-cultured and processed for immunofluorescence microscopy using antibodies specific to lamin A/C phosphorylated at serine 22 (LA/C pS22), antibody 1H11 against LAP2α, and DAPI to visualize DNA. Bar: 10 µm. (F) Isogenic LAP2α KO or WT clones were processed for western blotting using antibodies against lamin A/C phosphorylated at specific residues as indicated or a pan-lamin A/C antibody (E1). Western blot signals for phosphorylated lamins were quantified, normalized to total lamins A and C and expressed as fold difference to the WT samples (graphs on the right). Graphs display average fold difference ± S.E.M. $n_{WT}$ = 3, $n_{KO}$ = 3 (single data points are displayed), p values (paired student's t-test on log transformed values) are: lamin A pS392 p = 0.602, lamin C pS392 p = 0.493, lamin A pS22 p = 0.841, lamin C pS22 p = 0.826 (non-significant).

## Reduced lamin A/C mobility in the absence of LAP2α decreases chromatin diffusion

To address the relevance of mobile intranuclear lamin A structures for nuclear functions and the consequences upon its loss in LAP2α-deficient cells, we tested chromatin mobility, since intranuclear lamins and LAP2α were found to bind to chromatin in the nuclear interior (*Bronshtein et al., 2015*; *Gesson et al., 2016*). Chromatin shows anomalous sub-diffusion, which – unlike free (normal) diffusion – is a motion that is slowed by constraints, such as temporal binding to nucleoplasmic lamins (*Bronshtein et al., 2015*). Depletion of lamin A/C was shown to increase chromatin motion and to change the type of diffusion of telomeres, centromeres and genomic loci in the nuclear interior toward normal unrestricted diffusion (*Bronshtein et al., 2015*). To test how depletion of LAP2α affects chromatin motion, we expressed fluorescently labeled TRF1 (to detect telomeres) in mEos3.2-*Lmna* wildtype and LAP2α knockout cells and analyzed telomere trajectories for 20.5 min to determine the diffusion volume of telomeres (*Bronshtein et al., 2015*; *Vivante et al., 2019*). Strikingly, telomere motion was significantly reduced in the absence of LAP2α when compared to wildtype cells (*Figure 6A*, left panel, see also MSD plot in *Figure 6B*), and this effect was predominantly dependent on the presence of lamin A/C, since additional depletion of lamin A/C in LAP2α knockout cells (*Lmna*/*Lap2α* double knockout, see *Figure 6—figure supplement 1*) increased the telomere motion volume to levels similar to those observed in *Lmna* single knockout fibroblasts (*Figure 6A*, right panel). Thus, the absence of LAP2α and the resulting changes in nucleoplasmic lamin mobility toward a more stable form have direct functional consequences on chromatin diffusion in the nuclear interior, significantly reducing telomere motion.

Interestingly, chromatin movement was slightly, but significantly increased in *Lmna*/*Lap2α* double knockout cells when compared to *Lmna* single knockout fibroblasts (*Figure 6A*, right panel, Kruskal Wallis post test p<0.0001, effect size d = 0.3). This suggests that LAP2α may have additional lamin A/C-independent functions in the regulation of chromatin movement, which is in accordance with the broad, genome-wide interaction of LAP2α with euchromatin (*Gesson et al., 2016*). Thus, LAP2α may affect chromatin movement in the nuclear interior by at least two different means: (1) by regulating lamin A/C assembly and/or chromatin interaction, thereby affecting lamin-A-mediated chromatin regulation and (2) by binding to chromatin and causing a slight, but significant reduction in chromatin motion in the nuclear interior.

In summary, our data provide novel insight into the formation and regulation of nucleoplasmic lamin A/C by LAP2α (*Figure 7*). While the nucleoplasmic pool of lamin A/C is formed independently of LAP2α, the properties of lamin A/C in the nuclear interior are significantly affected by LAP2α in a lamin serine 22 phosphorylation-independent manner. Binding of LAP2α impairs formation of higher order structures of lamin A/C and/or their interaction with chromatin, thereby keeping them in a mobile and low assembly state, which in turn affects chromatin mobility.

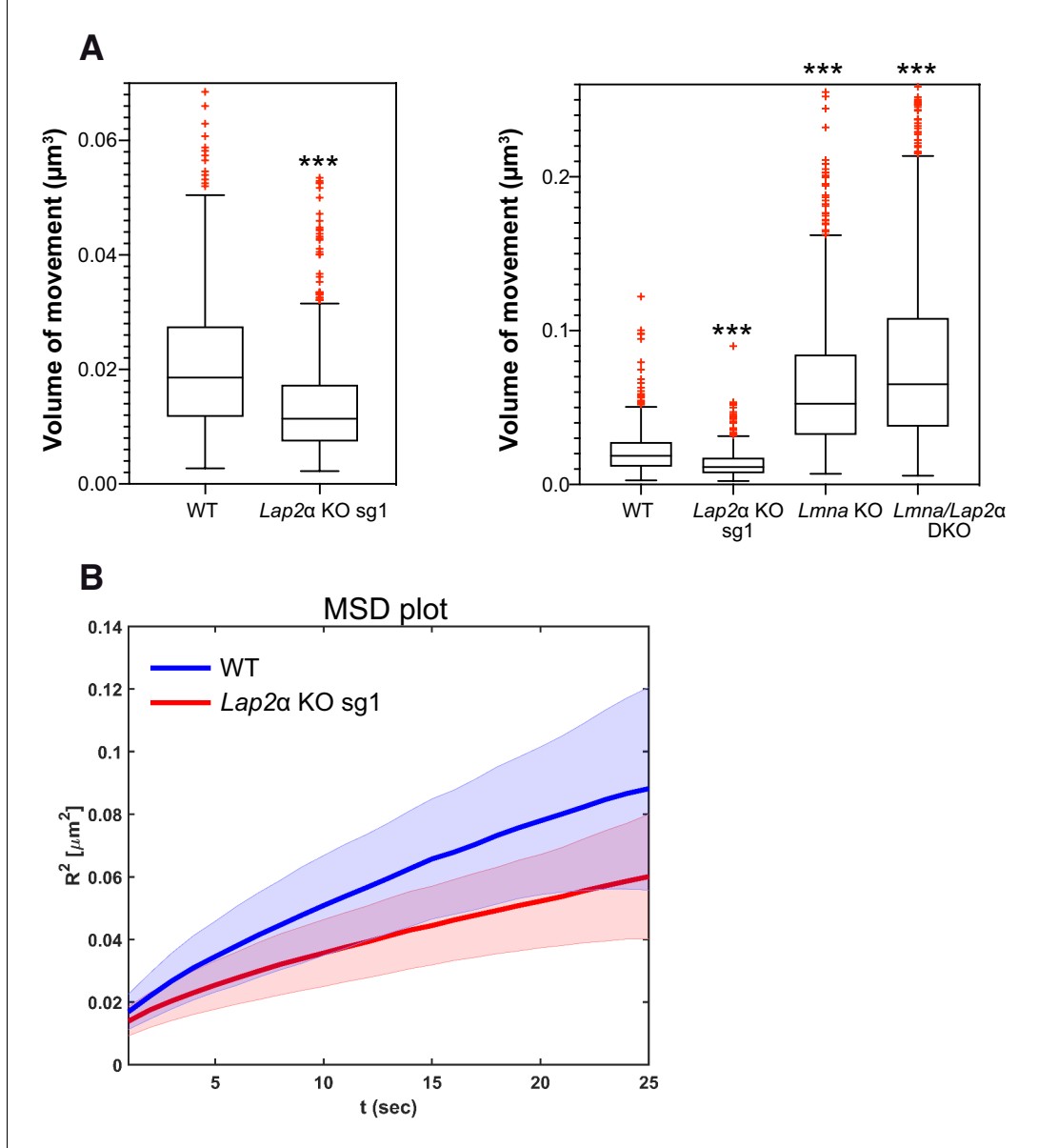

**Figure 6.** Lower lamin A/C mobility in the absence of LAP2α leads to a reduction in telomere movement. mEos3.2-*Lmna* WT or mEos3.2-*Lmna* fibroblasts lacking LAP2α (*Lap2α* KO sg1), lamin A/C (*Lmna* KO, see **Figure 6—figure supplement 1**), or both (*Lmna*/*Lap2α* DKO) were transiently transfected with a plasmid expressing DsRed-TRF1 to fluorescently label telomeres. The volume of telomere motion was then calculated based on its trajectory. (**A**) Box plots compare telomere motion between WT and *Lap2α* KO sg1 (left; ***p<0.0001 using Mann-Whitney U test, effect size d = 0.62) or between all four genotypes (right; Kruskal-Wallis test p<0.0001; post tests for WT ctrl versus each genotype are all highly significant with ***p<0.0001). The median is depicted within the first and third quartiles; whiskers: minimal and maximal datapoint excluding outliers (see also **Figure 6—source data 1**). (**B**) Mean square displacement (MSD) curves for telomere diffusion in mEos3.2-*Lmna* WT and LAP2α KO sg1 cells. The shaded area shows the distribution of 70% of the population. $n_{WT}$ = 471, $n_{Lap2\alpha KO}$=562, $n_{LmnaKO}$ = 925, $n_{DKO}$ = 1298.

The online version of this article includes the following source data and figure supplement(s) for figure 6:

**Source data 1.** Volume of telomere movement in WT, LAP2α KO, Lmna KO and DKO cells.

**Figure supplement 1.** Generation of isogenic *Lmna* knockout and *Lmna*/*Lap2α* double knockout mouse fibroblasts using CRISPR-Cas9.

## Discussion

Here, we show that LAP2α, a known binding partner of nucleoplasmic lamin A/C (**Dechat et al., 2000**; **Naetar and Foisner, 2009**) is essential to maintain lamins A and C in the nuclear interior in a mobile and low assembly state. This occurs in a lamin A/C S22 phosphorylation-independent

manner, likely by inhibiting lamin assembly through direct interaction of LAP2α with lamin A/C and/or by changing interactions of nucleoplasmic lamin complexes with chromatin.

Our study confirms and extends the finding that lamins A and C in the nuclear interior have fundamentally different properties compared to their counterparts at the nuclear periphery. While lamins at the nuclear periphery form stable 3.5-nm-wide filaments (*de Leeuw et al., 2018*; *Moir et al., 2000*; *Shimi et al., 2008*; *Turgay et al., 2017*), lamins in the nuclear interior are highly mobile (*Broers et al., 1999*; *Bronshtein et al., 2015*; *Shimi et al., 2008*) and easily extractable in salt and detergent-containing buffers (*Kolb et al., 2011*; *Naetar et al., 2008*). However, very little is known about the mechanisms securing these unique properties of lamins in the nuclear interior, given that in solution they have a strong drive to assemble and tend to form higher order structures (*Aebi et al., 1986*; *Moir et al., 1991*). Immunofluorescence microscopy has previously shown that intranuclear lamin A/C staining is significantly reduced in cells and tissues from LAP2α knockout mice (*Naetar et al., 2008*), leading to a model, where LAP2α is required for the formation and/or maintenance of the nucleoplasmic lamin pool (*Naetar and Foisner, 2009*). Based on data shown in this study, we propose a modified and updated version of the model suggesting that LAP2α is essential to maintain the highly mobile and soluble state of lamins in the nuclear interior. In the absence of LAP2α, lamins in the nucleoplasm are not lost but they are transformed into more stable, immobile structures (*Figure 7*) that can no longer be recognized by lamin A/C antibodies directed to an N-terminal epitope, which – in wildtype cells – favors nucleoplasmic over peripheral lamin staining (*Gesson et al., 2016*). These findings can be explained by the formation of higher order lamin complexes in the nuclear interior, whose properties may resemble more those of lamins at the nuclear periphery. This is supported by the increased resistance of nucleoplasmic lamins to extraction with detergent and salt-containing buffers and their decreased mobility in the absence of LAP2α. Moreover, high-resolution microscopy revealed significantly more and, importantly, larger extraction-resistant structures in LAP2α knockout versus wildtype cells. As for potential mechanisms describing how LAP2α can maintain lamins in a soluble and dynamic state, we can envisage at least two different not mutually exclusive mechanisms. Our in vitro data using purified recombinant LAP2α and lamin A protein suggest that direct binding of LAP2α to lamins may prevent them from assembling into higher order structures. However, one has to keep in mind that the in vitro assembly of lamins may not fully represent lamin assembly pathways within a cell, as previously indicated in a study using lamin-binding designed ankyrin repeat proteins (DARPins), where DARPins inhibiting in vitro lamin A assembly had no impact on incorporation of lamin A into the lamina in vivo and, vice versa, DARPins affecting the incorporation of lamin A into the nuclear lamina in vivo had no effect on lamin assembly in vitro. Nevertheless, we showed in a previous study that, unlike wildtype lamin A/C staining (*Naetar et al., 2008* and *Figure 3*), antibody-staining of the assembly-deficient lamin AΔK32 mutant in the nuclear interior is not affected in LAP2α knockout cells (*Pilat et al., 2013*). This suggests that nucleoplasmic lamin A assembly in the absence of LAP2α leads to the observed antibody epitope masking.

Besides affecting lamin assembly, LAP2α may also regulate the interaction of A-type lamin complexes with chromatin. In this model, the absence of LAP2α would lead to a tighter and more stable association of lamins with chromatin, which would in turn also lead to an increased extraction resistance and possibly decreased mobility. In support of this model, the post-fixation digestion of formaldehyde-fixed cells with DNase/RNase fully recovered the intranuclear lamin A/C staining using antibodies to the lamin A/C N-terminus, and chromatin motion was slowed down upon LAP2α knockout in a lamin A/C-dependent manner. However, the finding that the intranuclear lamin structures are resistant to 500 mM high salt, which is expected to extract chromatin-associated proteins (*Herrmann et al., 2017*), supports the notion that larger lamin A structures can be formed independently of chromatin. As these different scenarios are not mutually exclusive, we favor a model in which the absence of LAP2α induces the formation of higher order lamin A assemblies, which in turn may bind more tightly and stably to chromatin (*Figure 7*).

As phosphorylation of lamins provides another means to regulate lamin filament (dis)assembly in mitosis (*Heald and McKeon, 1990*; *Ward and Kirschner, 1990*) and to influence their assembly state, solubility and mobility in interphase (*Buxboim et al., 2014*; *Kochin et al., 2014*; *Moiseeva et al., 2016*), we tested the relationship between lamin phosphorylation and interaction with LAP2α. We found that LAP2α loss does neither affect total levels of lamins phosphorylated at serines 22 and 392 nor does it change the localization and solubility of phosphorylated lamins. Vice

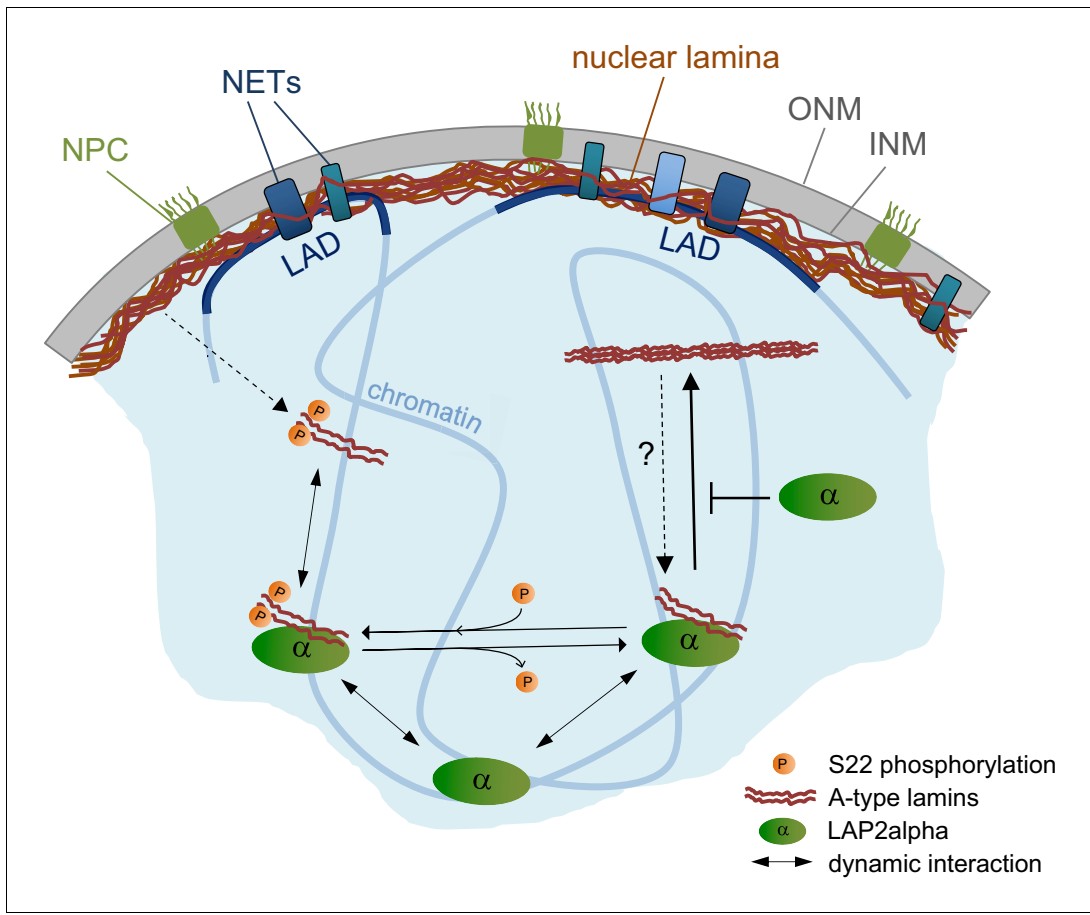

**Figure 7.** Model of LAP2α-dependent regulation of A-type lamins in the nucleoplasm. LAP2α dynamically interacts with lamins A and C in the nucleoplasm, independent of their phosphorylation status. While lamins at the periphery interact mainly with heterochromatic lamina-associated domains (LADs), nucleoplasmic A-type lamins and LAP2α bind to euchromatic genome regions regulating chromatin mobility. When hypophosphorylated lamins are not bound to LAP2α, they form larger, stable assemblies in the nucleoplasm. In the absence of LAP2α, these lamin A/C structures become more frequent, leading to altered, putatively less dynamic lamin A/C chromatin interaction and slower chromatin movement.

versa, LAP2α can interact with both phosphorylated and un(der)phosphorylated lamins, but it affects only the solubility and assembly state of un(der)phosphorylated lamins. Thus, interaction with LAP2α and lamin phosphorylation seem to be two independent, non-mutually exclusive mechanisms for keeping lamins in a low assembly state.

Surprisingly, the initial formation of the nucleoplasmic lamin A/C pool does not require LAP2α, as it can form normally from newly synthesized pre-lamin A and from soluble mitotic lamins in LAP2α knockout cells. Thus, we propose that these processes may primarily depend on other mechanisms, such as lamin A/C phosphorylation. At the onset of mitosis, lamins are targeted by cyclin-dependent kinase 1 (Cdk1) particularly at serines 22 and 392 in the N-terminal head and proximal to the C-terminal end of the rod, respectively (*Ward and Kirschner, 1990*), leading to a complete disassembly of the lamina (*Dechat et al., 2004*; *Heald and McKeon, 1990*; *Moir et al., 2000*). During post-mitotic nuclear envelope reassembly, lamins A and C are imported into the nucleoplasm of newly formed nuclei, followed by gradual dephosphorylation and re-assembly at the nuclear periphery (*Moir et al., 2000*; *Steen et al., 2003*; *Steen and Collas, 2001*). It is tempting to speculate that the majority of lamins A and C is still phosphorylated at S22 and possibly other residues when they are re-imported into the nucleoplasm after post-mitotic nuclear membrane reassembly, preventing their incorporation into the peripheral lamina. In the nucleoplasm, a fraction of the phosphorylated lamins may then bind to LAP2α. When lamins are gradually dephosphorylated during G1 (*Steen et al.,*

*2003*; *Steen and Collas, 2001*), LAP2α-bound lamin A/C may be maintained in a low assembly state, while free lamin A/C can assemble into the lamina at the periphery. Thus, in the absence of LAP2α, the nucleoplasmic lamin A/C pool may still form, but, whereas phosphorylated lamins stay soluble independently of LAP2α, hypophosphorylated lamins tend to form higher-order structures at the 'wrong' position, that is within the nucleoplasm as opposed to the nuclear periphery in LAP2α knockout cells. This model suggests that in wild-type cells, there are three pools of intranuclear A-type lamins (*Figure 7*): (1) lamins phosphorylated at S22 (and possibly other residues) that are soluble independent of LAP2α; (2) hypophosphorylated lamins bound to LAP2α that are soluble independent of their phosphorylation status; and (3) hypophosphorylated lamins not bound to LAP2α that tend to form immobile and extraction-resistant structures in the nuclear interior. Previous reports and our own data showing that a small fraction of nucleoplasmic A-type lamins displayed low mobility (*Bronshtein et al., 2015*; *Shimi et al., 2008*) in wildtype cells supports this model. In the absence of LAP2α, the balance between these different intranuclear lamin structures is disturbed toward increased levels of immobile, higher order lamin A/C structures.

What is the physiological relevance of a dynamic lamin A/C pool in the nucleoplasm and how is the nucleoplasmic lamin A/C pool regulated during various cellular processes? We propose that the unique dynamic properties allow intranuclear A-type lamins to fulfill a set of functions that is different from those covered by the peripheral lamina. One of the most prominent examples for different functions of the peripheral versus intranuclear lamins is their different role in chromatin organization. While the peripheral lamina is well known to mediate stable interaction with heterochromatic genomic regions, termed lamina-associated domains (LADs) (*Kind et al., 2013*; *van Steensel and Belmont, 2017*), thereby contributing to gene silencing in general (*Leemans et al., 2019*) or during differentiation (*Robson et al., 2016*), lamins in the nuclear interior have a much more complex role in regulating open chromatin and gene expression. Lamin A/C in the nuclear interior interact primarily with open euchromatic regions of the genome (*Gesson et al., 2016*; *Lund et al., 2015*) and affect their epigenetic landscape (*Bianchi et al., 2020*; *Briand et al., 2018*; *Cesarini et al., 2015*; *Oldenburg et al., 2017*). Furthermore, lamin A/C in the nucleoplasm has also been shown to directly interact with promoters and enhancers and seems to be involved in both positive and negative gene regulations (*Ikegami et al., 2020*; *Lund and Collas, 2013*; *Lund et al., 2013*). It is conceivable that gene regulatory pathways have to be highly flexible and dynamic to respond efficiently to internal and external cues and thus require a dynamic lamin complex that can dynamically associate with and regulate chromatin. Indeed, loss of LAP2α (*Gesson et al., 2016*) or disease-linked mutations in lamins (*Bianchi et al., 2020*; *Briand et al., 2018*; *Ikegami et al., 2020*; *Oldenburg et al., 2017*) affect epigenetic regulation and/or gene expression.

Another function of nucleoplasmic lamins is to regulate the mobility of chromatin in the three-dimensional nuclear space. Dynamic interactions of lamin complexes with telomeres and other genomic loci has been shown to limit free diffusion of chromatin inside the nucleus (*Bronshtein et al., 2015*). We demonstrate here that knockout of LAP2α reduces telomere motion mostly in a lamin A/C-dependent manner, most likely through the formation of more stable lamin A/C structures that bind chromatin more tightly and stably. It is reasonable to assume that a tightly regulated movement of genes inside the nucleus is important for controlling coordinated regulation of gene expression during various cellular processes. Notably, LAP2α may have another lamin A-independent role in the regulation of chromatin diffusion. By direct binding to chromatin, LAP2α may additionally slow down chromatin diffusion. However, the major contribution to restricted, anomalous chromatin diffusion in the nuclear interior seems to stem from nucleoplasmic lamin A/C, as its depletion has the highest impact on chromatin movement.

Overall, we speculate that a dynamic nucleoplasmic lamin A/C complex may be required to maintain genomic plasticity, allowing the cell to efficiently respond to external and internal cues, such as during cell differentiation. In line with this hypothesis, we found nucleoplasmic lamins and LAP2α in proliferating tissue progenitor cells in the epidermis, colon, skeletal muscle and in preadipocytes, while upon their differentiation (*Dorner et al., 2006*; *Gotic et al., 2010*; *Markiewicz et al., 2005*; *Naetar et al., 2008*) or cell cycle exit into a quiescence or senescence state (*Naetar et al., 2007*) LAP2α was downregulated at transcriptional and protein level leading to loss of the dynamic nucleoplasmic lamin A/C pool. Thus, when genome plasticity is no longer required, such es in terminally differentiated cells, dynamic lamin A/C levels may be reduced through downregulation of LAP2α. Accordingly, the absence of the dynamic nucleoplasmic lamin A/C pool in the LAP2α knockout

progenitor cells leads to impaired or delayed differentiation (*Gotic et al., 2010*; *Naetar et al., 2008*). This regulatory pathway may also be impaired in lamin-linked diseases, such as the premature aging disease Hutchinson Gilford Progeria Syndrome (HGPS), where LAP2α expression is lost, leading to premature senescence, while re-expression of LAP2α in HGPS cells rescues cell proliferation (*Vidak et al., 2015*).

In summary, we propose that lamin A/ C complexes in the nuclear interior are in a dynamic state, established by LAP2α, which is a prerequisite for their multiple roles in chromatin regulation both on a genome-wide scale and in a gene-specific manner during cell differentiation.

# Materials and methods

## Key resources table

| Reagent type (species) or resource | Designation | Source or reference | Identifiers | Additional information |
|---|---|---|---|---|
| Gene (*Mus musculus*) | *Lmna* | GenBank | Gene:ID_16905 | |
| Gene (*M. musculus*) | *Tmpo; Lap2* | GenBank | Gene:ID_21917 | |
| Gene (*Homo sapiens*) | *TMPO; LAP2* | GenBank | Gene:ID_7112 | |
| Strain, strain background (*E.c oli*) | NEB5-alpha Competent *E. coli* (high efficiency) | New England Biolabs | Cat# C2987 | Chemically competent |
| Strain, strain background (*E. coli*) | BL21-CodonPlus (DE3) | PMID:10562549 | | |
| Cell line (*M. musculus*) | mEos3.2-*Lmna* wildtype dermal fibroblasts | This paper | | See Materials and methods, section 'Generation and cultivation of cell lines' |
| Cell line (*M. musculus*) | mEos3.2- *Lmna* LAP2α knockout fibroblasts | This paper | | See Materials and methods, section 'Generation and cultivation of cell lines' |
| Cell line (*M. musculus*) | mEos3.2- *Lmna* lamin A/C knockout fibroblasts | This paper | | See Materials and methods, section 'Generation and cultivation of cell lines' |
| Cell line (*M. musculus*) | mEos3.2- *Lmna* LAP2α-lamin A/C double knockout fibroblasts | This paper | | See Materials and methods, section 'Generation and cultivation of cell lines' |
| Cell line (*H. sapiens*) | LAP2α knockout HeLa cells | This paper | | See Materials and methods, section 'Generation and cultivation of cell lines' |
| Transfected construct (*M. musculus*) | *Lmna*-targeting construct | This paper | | See Materials and methods, section 'Generation of *Lmna*-specific targeting construct' |
| Transfected construct (*M. musculus* and *H. sapiens*) | sgRNAs targeting *Lmna* or *Lap2* in pX458 | This paper | | See Materials and methods, section 'Generation and cultivation of cell lines' |
| Antibody | Anti-LAP2α (rabbit, polyclonal) | PMID:11864981 | | WB (1:5000) IF (1:800) |
| Antibody | Anti-LAP2α 1H11 (mouse, monoclonal) | Max Perutz Labs Monoclonal Antibody facility, see also PMID:26798136 | | WB (1:100) IF (1:30) |
| Antibody | Anti-LAP2α Ab15-2 (mouse, monoclonal) | PMID:9707448 | | WB (undiluted hybridoma supernatant) |

*Continued on next page*

*Continued*

| Reagent type (species) or resource | Designation | Source or reference | Identifiers | Additional information |
|---|---|---|---|---|
| Antibody | Anti-LAP2 Ab12 (mouse, monoclonal) | PMID:9707448 | | WB (1:4) |
| Antibody | Anti-lamin A/C 3A6 (mouse, monoclonal) | Max Perutz Labs Monoclonal Antibody facility, see also PMID:26798136 | | WB (1:50) IF (1:50) |
| Antibody | Anti-lamin A/C E1 (mouse, monoclonal) | Santa Cruz Biotechnology | Cat# sc-376248, RRID: AB_10991536 | WB (1:1000) IF (1:100) |
| Antibody | Anti-lamin A/C N18 (goat, polyclonal) | Santa Cruz Biotechnology | Cat# sc-6215, RRID:AB_648152 | WB (1:1000) IF (1:100) |
| Antibody | Anti-lamin A/C pS22 (rabbit polyclonal) | Invitrogen/ Thermo Fisher Scientific | Cat# PA5-17113, RRID: AB_10989809 | WB (1:200) IF (1:50) |
| Antibody | Anti-lamin A/C pS22 D2B2E (rabbit, monoclonal) | Cell Signaling | Cat# 13448, RRID:AB_2798221 | IP (5 µl/IP) |
| Antibody | Anti-lamin A/C pS392 (rabbit, polyclonal) | Abcam | Cat# ab58528, RRID:AB_883054 | WB (1:500) |
| Antibody | Anti-lamin B1 (rabbit, polyclonal) | Proteintech | Cat# 12987–1-AP, RRID: AB_2136290 | WB (1:500) IF (1:100) |
| Antibody | Anti-mEos2 (rabbit polyclonal) | Badrilla | Cat# A010-mEOS2, RRID: AB_2773027 | IF (1:200) |
| Antibody | Anti-γ-tubulin GTU88 (mouse, monoclonal) | Sigma Aldrich | Cat# T6557, RRID:AB_477584 | WB (1:5000) |
| Antibody | Anti-actin I-19 (goat, polyclonal) | Santa Cruz Biotechnology | Cat# sc-1616, RRID:AB_630836 | WB (1:500) |
| Recombinant DNA reagent | pSpCas9(BB)—2A-GFP; pX458 (plasmid) | Addgene | RRID:Addgene_48138 | sgRNAs targeting *Lmna* or *Lap2* were inserted into this vector |
| Recombinant DNA reagent | pSpCas9(BB)—2A-mCherry (plasmid) | This paper | | See Materials and methods, section 'Vectors' |
| Recombinant DNA reagent | pEGFP-myc-Lamin A (plasmid) | This paper | | See Materials and methods, section 'Vectors' |
| Recombinant DNA reagent | dsRed-TRF1 (plasmid) | PMID:26299252 | | |
| Sequence-based reagent | h*LAP2α* sgRNA1 | This paper | sgRNA | AGTTCGGTACTGCCCAAAGG |
| Sequence-based reagent | h*LAP2α* sgRNA2 | This paper | sgRNA | GGAACTACTCCCTCTGGTGG |
| Sequence-based reagent | m*Lmna* sgRNA1 | This paper | sgRNA | CACTCGGATCACCCGGCTGC |
| Sequence-based reagent | m*Lmna* sgRNA2 | This paper | sgRNA | ACGCACGCGATCGATGTACA |
| Sequence-based reagent | m*Lap2α* sgRNA1 | This paper | sgRNA | AGTTCAGTCCTGCCCAAAGG |
| Sequence-based reagent | m*Lap2α* sgRNA3 | This paper | sgRNA | CAAGAAAGTGAAGTCCGCTA |

*Continued on next page*

*Continued*

| Reagent type (species) or resource | Designation | Source or reference | Identifiers | Additional information |
|---|---|---|---|---|
| Peptide, recombinant protein | Q5 High Fidelity DNA polymerase | New England Biolabs | Cat# M0491 | |
| Commercial assay or kit | RevertAid First Strand cDNA synthesis kit | Thermo Fisher Scientific | Cat# K1622 | |
| Commercial assay or kit | KAPA SYBR Green 2x PCR master mix | Kapa Biosystems | Cat# KK4602 | |
| Commercial assay or kit | Gibson assembly master mix | New England Biolabs | Cat# E2611 | |
| Commercial assay or kit | GoTaq green master mix | Promega | Cat# M7122 | |
| Software, algorithm | TIDE | PMID:25300484 | | https://tide.nki.nl/ |
| Software, algorithm | Visiview 4.4 software | Visitron systems | | |
| Software, algorithm | FIJI | Open source | RRID:SCR_002285 | https://imagej.net/Fiji |
| Software, algorithm | Zeiss ZEN 2.1 software | Zeiss | | |
| Software, algorithm | Matlab | MathWorks | RRID:SCR_001622 | |
| Software, algorithm | SymPho Time software | PicoQuant | | |

## Generation and cultivation of cell lines

To generate mEos3.2-*Lmna* mouse dermal fibroblast cell lines, immortalized fibroblasts derived from the back skin of wild-type and LAP2α knockout littermates (*Naetar et al., 2008*) were transfected with the vector pSpCas9(BB)−2A-GFP (pX458, plasmid #48138 from Addgene, Watertown, MA) (*Ran et al., 2013*) carrying m*Lmna*-specific sgRNA1 (see Key Resource Table) and Cas9 from *S. pyogenes* with 2A-EGFP, and a *Lmna*-specific targeting construct, where the fluorophore mEos3.2 (*Zhang et al., 2012*) was inserted in frame into exon 1 of the *Lmna* gene in front of the first codon (*Figure 2A*, see also 'Generation of *Lmna*-specific targeting construct'). Cells were sorted for EGFP expression using a FACS Aria Illu (Becton Dickinson, Franklin Lakes, NJ), cultivated for 14 days, followed by a single cell sort for mEos3.2-expressing cells. Single cell clones were outgrown and further characterized by genotyping PCR, long-range PCR, sequencing of the targeted locus and Western blotting.

The wildtype mEos3.2-*Lmna* clone #21 was selected to create isogenic *Lap2α* knockout clones, as well as *Lmna* knockout and *Lmna/Lap2α* double knockout clones. Cells were transfected with the vector pSpCas9(BB)−2A-mCherry (modified pSpCas9(BB)−2A-GFP, see also 'Vectors') carrying m*Lap2α*-specific sgRNA1 or 3, or m*Lmna*-specific sgRNA2 (see Key Resource Table) and Cas 9 from *S. pyogenes* with 2A-mCherry. mCherry-positive single cells were FACS-sorted and knockout clones were identified by western blot. Knockout clones and wildtype control clones were further characterized by sequencing of a PCR product derived from isolated genomic DNA spanning the expected Cas9 cut site (primers m*Lap2α*-f and -r, or m*Lmna*-f and -r: see 'Primers'). Sequences were analyzed using the TIDE software available online (https://tide.nki.nl/) (*Brinkman et al., 2014*). Absence of continuous Cas9 expression was verified by mCherry FACS analysis using an LSR Fortessa (Becton Dickinson).

To generate LAP2α knockout HeLa cell clones, HeLa cells were transfected with pSpCas9(BB)−2A-GFP carrying h*LAP2α*-specific sgRNA1 or 2 (see Key Resource Table). GFP-positive cells were sorted and plated for generation of single cell clones. LAP2α knockout clones were identified by western blotting. Knockout clones and wildtype control clones were further characterized by sequencing of a PCR product spanning the expected Cas9 cut site (primers h*LAP2α*-f and -r: see 'Primers'). Sequences were analyzed using TIDE. Absence of continuous Cas9 expression was verified by GFP FACS analysis.

All cells were maintained at 37°C and 5% $CO_2$ in Dulbecco's modified Eagle's medium (DMEM) supplemented with 10% fetal calf serum (FCS), 2 mM glutamine, 50 U/ml penicillin and 50 µg/ml streptomycin (P/S) (all from Sigma-Aldrich, St. Louis, MO). Non-essential amino acids (PAN-Biotech, Aidenbach, Germany) were routinely added for mouse dermal fibroblasts. After sorting, cells were kept in medium additionally containing Antimycotic-Antibiotic (Gibco/Thermo Fisher Scientific, Waltham, MA) and Plasmocin (25 µg/ml, Invivogen, San Diego, CA) to prevent sorter-induced contamination. Cells were routinely inspected for mycoplasma contamination after staining nuclei with DAPI or Hoechst dye (Thermo Fisher Scientific). Additionally, especially if mycoplasma contamination was suspected based on DAPI staining, cells were tested for mycoplasma contamination using the MycoAlert Mycoplasma detection kit (Lonza, Basel, Switzerland). To determine cell viability, cells were lysed in CASYblue (OLS OMNI Life Science, Bremen, Germany), containing MeOH. The lysed cell sample was measured using a Casy Counter (OLS OMNI Life Science) and the cell volume profile was used to define the volume cut-off for dead cells. Living cell samples were then measured according to manufacturer's instructions using these settings to determine the percentage of viable cells within the sample.

## Live cell imaging

Cells were plated on 35 mm glass-bottom µ-dishes (Ibidi, Gräfelfing, Germany) in high-glucose phenol-red free DMEM (Gibco Fluorobrite) supplemented with 10% FCS, L-glutamine, and P/S. For mEos3.2 cells, dishes were pre-coated with Cell-tak (Corning, Corning, NY) according to manufacturer's instructions. HeLa cells were transfected with plasmids expressing EGFP-tagged pre-lamin A, mature lamin A, pre-lamin A delK32 or mature lamin A delK32 (see'Vectors') using nanofectin (PAA Laboratories, Toronto, Canada) according to manufacturer's instructions. To avoid substantial overexpression, a maximum of 1 µg plasmid DNA per 35 mm dish was used. Cells were imaged 5 hr or 24 hr post transfection.

Imaging was performed using a Visitron Spinning disc microscope (Zeiss, Oberkochen, Germany) under controlled environmental conditions at 37°C and 5% $CO_2$, using a Plan-Apochromat 63x/1.4 Oil DIC III objective and an EM-CCD camera. Images (Z stacks) were obtained by automated multi-positioning (40 positions per 35 mm dish) every 20 min with the same excitation strength and exposure time using Visiview 4.4 software (Visitron Systems, Puchheim, Germany). Images were processed in FIJI. For experiments with HeLa cells, all time-lapse images were smoothed by 1-pixel radius Gaussian Blur filter, the frame was cropped to fit the cells of interest and substacks were created from Z stacks to include the focal plane of appropriate time points starting with the beginning of mitosis until late G1. These substacks were then used to create short movies and the images presented in the results section. The data were quantified manually by drawing a three pixel wide bar across the nuclei of selected cells and creating plot profiles for each cell. The peripheral (P) lamin signal was calculated by averaging two peak measurements, one from each end of the drawn line, and the nucleoplasmic (N) intensity as the average of 30 measurements spanning the center of the drawn line, followed by calculation of the N/P ratio.

Automated quantification of the N/P ratio for mEos3.2-lamin A/C cells was done using custom-made FIJI plugins, where mitotic cells are manually extracted from the images, followed by automatic analysis of time series stacks and data visualization of nucleoplasmic to peripheral signal intensity ratio over time. Specifically, frames, slices and fields of view with relevant cells were manually selected and extracted for automated analysis. Nuclei were segmented by thresholding, marking regions of interest. These regions were further reduced and subtracted to define nucleoplasm and nuclear periphery. Average intensity values of mEos3.2 were extracted from these regions and the signal ratio (nucleoplasm/periphery) was calculated.

## Immunofluorescence and cell cycle staging using DAPI

For standard immunofluorescence (IF), cells were seeded on uncoated glass coverslips (1.5H, Marienfeld-Superior, Lauda-Königshofen, Germany) and fixed with 4% paraformaldehyde for 10 min at room temperature. To stop fixation and permeabilize the cells, coverslips were incubated in PBS with 0.1% Triton X-100 and 50 mM $NH_4Cl$, followed by incubation in primary and secondary antibody (for a list of all antibodies see Key Resource Table). To reverse chromatin-dependent epitope masking of nucleoplasmic lamins, cells were treated with 100 µg/ml DNase I and 100 µg/ml RNase A

for 30 min at room temperature. DNA was stained with DAPI (0.5 µg/ml) and cells were mounted using Vectashield (Vector Laboratories, Burlingame, CA). Immunofluorescence slides were routinely imaged using an LSM710 confocal microscope (Zeiss) equipped with a Plan-Apochromat 63x/1.4 Oil DIC M27 objective and standard photomultiplier tubes (PMTs) for sequential detection, as well as an Airyscan detector for high-resolution imaging. Image acquisition was done using Zeiss ZEN 2.1 software, followed by image processing using FIJI, including adjustment of the digital offset for high resolution Airy scan images to avoid over- and undersaturation of the fluorescent signal. IF images with standard resolution were smoothened by 1-pixel radius Gaussian Blur filter. Extraction-resistant lamin A/C structures were quantified using a custom-made FIJI plugin, where cells were first defined using the FIJI built-in auto-threshold function on average Z projections of mEos3.2-lamin A/C, followed by identification and quantification (number and area) of intranuclear structures using the threshold function with the built-in 'Huang' algorithm, followed by the particle analyzer function, including particles between 0.0001 and 5 µm, but excluding a rim of 0.8 µm from the nuclear periphery to avoid counting of signals within the nuclear lamina.

For DAPI-based cell cycle staging and combined determination of nucleoplasmic to peripheral ratio of mEos3.2-tagged lamin A/C, cells were imaged using the Visitron Spinning disc microscope (Zeiss) and a custom-made slide scanner, allowing the automated acquisition of multiple image stacks (400 fields of view/sample). Images were analyzed using a custom-made FIJI plugin, where the cell cycle stage of each cell was determined based on the DAPI intensity and a cell cycle profile was created. Initial regions of interest were extracted by segmenting nuclei using thresholding and watersheding. DAPI intensity was measured by summing the integrated density of DAPI signals within regions of interest. Initial regions of interest were further reduced and subtracted to define nucleoplasm and nuclear periphery. Intensity of the nucleoplasm was measured by taking the median value of the minimum projection of the regions of interest. Intensity of the nuclear periphery was measured by taking the upper quartile of the maximum projection of the regions of interest. The nucleoplasmic/peripheral lamin A/C ratio of the cells was calculated and plotted against the cell cycle stage.

## Continuous photobleaching and fluorescence correlation spectroscopy

Continuous photobleaching was performed using a confocal microscope (Leica TCS SP8 SMD, Leica Microsystems, Wetzlar, Germany). For illumination, the PicoQuant Picosecond Pulsed Diode Laser Head of 40 MHz 470 nm was used (laser intensity was set to ~1 µW). The light is focused onto a small confocal volume through a 63x water immersion objective lens with NA = 1.2 (Leica HC PL APO 63x/1.20 W CORR). The signal is detected by sensitive detectors (Leica HyD SMD) within the detection window (500–600 nm). A specific point was chosen in the nuclear interior and a 'point measurement' was performed for measuring the intensity at a high frequency (~1 KHz) for approximately 60 s per cell. CP data analysis was performed with Matlab and a custom-made algorithm (*Bronshtein et al., 2015*).

Fluorescence correlation spectroscopy (FCS) curves were extracted from the region where the fluorescence intensity is constant (approximately 30 s after beginning of the measurement) in order to avoid inaccuracy due to bleaching. The FCS analysis was done by SymPho Time software (Pico-Quant, Berlin, Germany). The best fit was achieved with the FCS Triplet 3D fitting model:

$$G(t) = \left[1 + \sum_{j=0}^{n_{Trip}-1} T[j] \left[\exp\left(-\frac{t}{\tau_{Trip}[j]}\right) - 1\right]\right] \sum_{i=0}^{n_{Diff}-1} \frac{\rho[i]}{\left[1 + \left[\frac{t}{\tau_{Diff}[i]}\right]^{a[i]}\right] \left[1 + \frac{\left[\frac{t}{\tau_{Diff}[i]}\right]^{a[i]}}{k^2}\right]^{0.5}} + G_{Inf}$$

$$w_0 = \left[\frac{V_{Eff}}{k}\right]^{\frac{1}{3}} \pi^{-0.5}$$

$$D[k] = \frac{w_0^2}{4\tau_{Diff}[k]}$$

## Model parameters

Number of triplet (dark) states $n_{Trip} = 1$
Number of independently diffusing species $n_{Diff} = 2$
Effective excitation volume $V_{Eff} = 0.267\,[fl]$
Length to diameter ratio of the focal volume $k = 7.92$
Anomaly parameter of the $i^{th}$ diffusing species $a_{1,2} = 1$

## Fitting parameters

Contribution of the $i^{th}$ diffusing species - $\rho[i]$
Diffusion time of the $i^{th}$ diffusing species - $\tau_{Diff}[i]$
Dark (triplet) fractions of molecules - $T$
Lifetime of the dark (triplet) states - $\tau_{Trip}$
Correlation offset - $G_{Inf}$
Effective lateral focal radius at $\frac{1}{e^2}$ intensity - $w_0$
Diffusion constant of the $k^{th}$ diffusing species - $D[k]$

To explain the observed differences in the measured diffusion of free lamin A complexes between wild-type and LAP2α knockout cells, the diffusion was modeled making several basic assumptions on protein shape and structure, based on the Stokes-Einstein relation for normal diffusion:

$$D = \frac{\kappa_B T}{6\pi\eta\alpha}$$

where $k_B$ is the Boltzmann constant, $T$ is the temperature, $\eta$ is the nucleoplasm viscosity and $\alpha$ is the protein radius, assuming a spherical shape.

$$D \propto \frac{1}{\alpha}$$

and one can assume that the protein volume is proportional to its molecular weight. (**Erickson, 2009**).

Therefore, $\frac{D_{WT}}{D_{KO}} = \frac{\alpha_{KO}}{\alpha_{WT}} = \sqrt[3]{\frac{M_{KO}}{M_{WT}}}$, and the ratio of the FCS diffusion rates $D_{WT}/D_{KO}$ should equal the cube root of the inverted nucleoplasmic lamin complex molecular weight ratio $M_{KO}/M_{WT}$, allowing to make conclusions on the relative molecular weight of measured complexes in WT and KO cells.

### Biochemical extraction experiments

Cells were washed twice with PBS (Sigma-Aldrich) and directly scraped off the plate in cold extraction buffer on ice (20 mM Tris-HCl pH7.5, 150 mM NaCl, 2 mM EGTA, 2 mM MgCl₂, 0.5% NP-40, 1 mM DTT, 1 U/ml Benzonase from Novagen/EMD Millipore, Temecula, CA, 1x Complete Protease Inhibitor Cocktail from Sigma-Aldrich, 1x Phosphatase inhibitor cocktail 2 and 3 from Sigma-Aldrich). For mouse fibroblasts, extra NaCl was added to a final concentration of 500 mM. Extracts were incubated for 10 min on ice and then centrifuged for 10 min at 4000 rpm in a Megafuge 1.0R (Haereus, Hanau, Germany). Pellets were resuspended in equal volumes of extraction buffer and sonicated with a Bandelin Sonopuls HD200 sonicator (Bandelin, Berlin, Germany; settings: MS73/D at 50% intensity) for 3 s to solubilize insoluble material. Total, supernatant and pellet fractions were analyzed by immunoblotting.

For in situ extraction of mEos3.2-lamin A/C mouse fibroblasts, cells were grown on glass coverslips (1.5H, Marienfeld) that were pre-coated with Cell-tak (Corning) according to manufacturer's instructions to avoid detachment of cells due to extraction. Cells were washed twice with PBS and incubated for 10 min in extraction buffer containing 500 mM NaCl (without benzonase), followed by fixation in 4% paraformaldehyde according to the routine IF protocol (see 'Immunofluorescence and cell cycle staging using DAPI'). To amplify the mEos3.2 signal, cells were stained with an anti-mEos2 antibody (Badrilla, Leeds, UK) following the standard IF protocol.

## Recombinant protein expression and sedimentation assays

Recombinant full-length human LAP2α and the truncation mutant LAP2α$_{1-414}$ were expressed in *E. coli* BL21 (DE3) using the plasmid pET23a(+) (*Vlcek et al., 1999*), which adds a 6x histidine-tag to the C-terminus of the expressed proteins for further affinity purification. Protein expression was induced in bacterial cultures at an OD$_{600nm}$ of 0.6–0.7 with 0.5 mM Isopropyl-β-D-thiogalactopyrano-side (IPTG) for 3 hr. Bacterial pellets were then resuspended in 10 mM Tris-HCl pH 8.0, 100 mM NaCl, and 1 mM DTT, followed by cell lysis in the presence of 1x protease inhibitor cocktail (Sigma Aldrich). 5 µg/ml DNase I and 10 µg/ml RNase A were added and inclusion bodies of the respective recombinant proteins were pelleted by centrifugation. Poly-histidine-tagged recombinant proteins were purified using Ni-NTA-Agarose (Qiagen, Hilden, Germany) beads according to manufacturer's instructions and were analyzed by SDS-polyacrylamide gel electrophoresis (SDS-PAGE). Pure protein fractions were dialyzed twice against 8 M Urea, 10 mM Tris-HCl pH 8.0, 300 mM NaCl and 1 mM DTT using a cut-off of 12–14 kDa to remove Imidazole and ß-Mercaptoethanol. Purified recombinant human wild-type lamin A was a kind gift of Prof. Harald Herrmann, DKFZ, Heidelberg, Germany.

For sucrose density gradient centrifugation, recombinant proteins were dialyzed against 10 mM Tris-HCl pH 7.5, 1 mM DTT, 300 mM NaCl and 10% sucrose and were layered on top of a sucrose gradient ranging from 10% to 30% atop a 70% sucrose cushion in a centrifuge tube. Samples were centrifuged for 20 hr at 190,000 x g in a SW-40 Rotor (Beckman Optima L-70 or Beckman Optima L-80 XP, Beckman Coulter, Brea, CA) at 4°C. Collection of the sedimented protein layers was done by pipetting off fractions from the top to the bottom of the gradient, starting with the applied dia-lyzed protein sample (supernatant). Aliquots of all gradient fractions were analyzed on an SDS-poly-acrylamide gel stained with Gel CodeTM Blue Safe Protein Stain (Thermo Fisher Scientific) and density measurement of the protein bands was done with ImageJ software.

To induce lamin assembly, recombinant proteins were dialyzed stepwise against 10 mM Tris-HCl pH 7.5, 1 mM DTT containing decreasing salt concentrations, starting at 300 mM NaCl, followed by 200 mM NaCl, and ending at 100 mM NaCl. Each step was done at room temperature for 30 min. Samples were then analyzed by centrifugation at 13,000 rpm for 10 min (Eppendorf table top centri-fuge, Eppendorf, Hamburg, Germany) at room temperature. Supernatant and pellet fractions were analyzed by SDS-PAGE as described above for sucrose gradient fractions.

## Immunoprecipitation and immunoblotting

For co-immunoprecipitation (IP), cells were scraped off the plate in IP buffer containing 20 mM Tris-HCl pH7.5, 150 mM NaCl, 2 mM EGTA, 2 mM MgCl$_2$, 0.5% NP-40, 1 mM DTT, 1 U/ml Benzonase (Novagen), 1x Complete Protease Inhibitor cocktail (Sigma-Aldrich) and Phosphatase inhibitor cock-tail 2 and 3 (Sigma-Aldrich), and incubated 10 min on ice. The soluble fraction after centrifugation for 10 min at 4000 rpm in a Megafuge 1.0R (Haereus) was used as input for the IP (1 mg/IP). Incuba-tion with antibody (5 µg/IP) was done overnight at 4°C, followed by incubation with BSA-blocked proteinA/G dynabeads (Pierce/Thermo Fisher Scientific) for 4 hr at 4°C. Beads were washed three times in IP buffer without benzonase, followed by elution of complexes from the beads using SDS PAGE sample buffer. Samples were analyzed by SDS PAGE and immunoblotting.

Immunoblots were treated with primary antibodies (see Key Resource Table) overnight and with horse radish peroxidase-coupled secondary antibodies (Jackson ImmunoResearch, Westgrove, PA) for 1 hr at room temperature. Signal detection was done using SuperSignal West Pico plus chemilu-minescent substrate (Pierce/Thermo Fisher Scientific) and the ChemiDoc Gel Imaging system (Bio-Rad, Hercules, CA). Image analysis and quantification was done using the Image Lab software (Bio-Rad).

## Telomere tracking

Cells were transfected with the DsRed-TRF1 plasmid (*Bronshtein et al., 2015*) one day before imag-ing. For imaging, cells were placed in an incubator (Tokai, Shizuoka-Ken, Japan) mounted on an inverted Olympus IX-81 fluorescence microscope coupled to a FV-1000 confocal set-up (Olympus, Tokyo, Japan) using a UPLSAPO 60X objective lens with a numerical aperture of 1.35. Each nucleus was measured 50 times in three dimensions (3D) for a total time of 20.5 min. Imaris image analysis software package (Bitplane, Zurich, Switzerland) was used for correcting the nucleus drift and rota-tion and for identifying the coordinates of labeled telomeres. Only telomeres that were tracked over

all the 50 time points were considered for further data analysis. For calculating the volume covered by each telomere during its whole motion, we used the Convex hull algorithm using a custom-made Matlab code.

## Quantitative real-time PCR for detecting untagged and mEos3.2-tagged lamin A/C mRNA levels

RNA was isolated from mEos3.2-lamin A/C WT#21 cells using the RNeasy mini plus kit (Qiagen) according to manufacturer's instructions. cDNA was synthesized from 500 ng total RNA using the RevertAid First Strand cDNA synthesis kit (Thermo Fisher Scientific) and analyzed by qPCR using the KAPA SYBR Green 2x PCR master mix (Kapa Biosystems, Wilmington, MA) in an Eppendorf Realplex 2 Mastercycler according to manufacturer's instructions. Primers specific for untagged and tagged lamins A/C (wt LAC-f and -r; mEos LA/C-f and -r – see 'Primers') were used to generate PCR products that were gel-extracted and used as a template in real time PCR to generate a standard curve (DNA concentration versus threshold cycle). WT#21 cDNA was then analyzed by real-time PCR using the same primers and DNA/RNA concentration of the specific template was calculated from the standard curve.

## Generation of *Lmna*-specific targeting construct

The *Lmna*-specific targeting construct was assembled from four fragments that were amplified by PCR using the following primer pairs and templates (see also 'Primers'):

| Fragment | F primer (5'- 3') | R primer (5'- 3') | Template |
|---|---|---|---|
| 1 | m*Lmna*-Frag1-f | m*Lmna*-Frag1-r | Mouse genomic DNA |
| 2 | mEos-Frag2-f | mEos-Frag2-r | mEos3.2-C1 (Addgene, plasmid #54550) |
| 3 | m*Lmna*-Frag3-f | m*Lmna*-Frag3-r | Mouse genomic DNA |
| 4 | m*Lmna*-Frag4-f | m*Lmna*-Frag4-r | Mouse genomic DNA |

Primers were designed to generate overlapping fragments that were assembled using the Gibson assembly master mix (New England Biolabs, Ipswich, MA) according to manufacturer's instructions. The vector pUC18 (*Norrander et al., 1983*) was digested with SalI and EcoRI (New England Biolabs) creating vector ends overlapping with fragments 1 and 4, respectively, and added to the Gibson assembly reaction. The final *Lmna* targeting construct in pUC18 was sequence-verified and contained *Lmna* exon one and its flanking non-coding sequences, where mEos3.2 was inserted in frame into exon one before the first codon (removing the start codon). An additional EcoRI site was inserted directly after mEos3.2 and the recognition sequence of *Lmna*-specific sgRNA1 was altered within exon 1 (5' CACTCGGATCACCCGcCTaC 3', mutations are indicated in lowercase) to avoid recutting and potential creation of Indels in the modified allele. The *Lmna* targeting construct in pUC18 was amplified in high-efficiency NEB5-alpha competent *E. coli* (New England Biolabs).

## Genotyping PCR and long range (LR) PCR for modified mEos3.2-*Lmna* knock-in allele

To identify clones with a modified mEos3.2-*Lmna* knock-in allele, genomic DNA was isolated from single-cell clones using the QuickExtract DNA extraction solution (Epicentre/Lucigen, Middleton, WI) according to manufacturer's instructions and analyzed for the presence of the wild-type and knock-in *Lmna* allele by PCR using the GoTaq green master mix (Promega, Madison, WI) and the following primer pairs (see also 'Primers'):

| PCR | Forward primer (5'- 3') | Reverse primer (5'- 3') | Product |
|---|---|---|---|
| WT | m*Lmna*-WT-f | m*Lmna*-WT-r | 213 bp (KI: 894 bp) |
| Knock-in | m*Lmna*-KI-f | m*Lmna*-KI-r | 250 bp |

LR-PCR was performed in clones with at least one knock-in allele to verify correct integration of the construct at the 3-prime and 5-prime side with one primer of each pair outside the homology

region of the *Lmna* targeting construct (see also *Figure 2—figure supplement 1*). PCR reactions were set up using the Q5 DNA polymerase (New England Biolabs) according to manufacturer's instructions and the following primer pairs:

| PCR | Forward primer (5'- 3') | Reverse primer (5'- 3') | Product |
|---|---|---|---|
| 5-prime | *Lmna*-LR-5'-f | *Lmna*-LR-5'-r | 2566 bp |
| 3-prime | *Lmna*-LR-3'-f | *Lmna*-LR-3'-r | 1710b p |

## Vectors

pSpCas9(BB)−2A-mCherry was generated from pSpCas9(BB)−2A-GFP (Addgene, plasmid #48138) by removing EGFP and the T2A sequence via EcoRI digestion. The final vector was then assembled using the Gibson assembly master mix (New England Biolabs) and two overlapping fragments containing the T2A sequence and mCherry generated by PCR using the following primers and templates (see 'Primers'):

| Fragment | Forward primer (5'- 3') | Reverse primer (5'- 3') | Template |
|---|---|---|---|
| T2A | T2A-f | T2A-r | pSpCas9(BB)−2A-GFP |
| mCherry | mCherry-f | mCherry-r | pLVX mCherry (Takara Bio) |

Vectors expressing different lamin A constructs were all derived from pEGFP-myc-LMNA (*Moir et al., 2000*). pEGFP-myc-Lamin A del K32, harboring a deletion of lysine at position 32 (*Bertrand et al., 2012*), was derived from pEGFP-myc-LMNA by site directed mutagenesis as previously described (*Pilat et al., 2013*). Wildtype and delK32 mature lamin A constructs were generated by PCR from pEGFP-myc-LMNA and pEGFP-myc-Lamin A delK32, respectively, using primers hLaminA-NheI-f and hLaminA-NheI-r.

## Primers

| Primer | Sequence (5' - 3') |
|---|---|
| h*LAP2α*-f | ACCTCAGGGCAACTTTAAAGCAA |
| h*LAP2α*-r | TCTACATCCAGTGGGGGCATA |
| m*Lap2α*-f | GGGTCTTTTATGGGCCATTTTTGT |
| m*Lap2α*-r | CTCTTCCCTCCACGGCAAA |
| m*Lmna*-f | ACCCCCTCCCTTCTATGTCC |
| m*Lmna*-r | GGAAGTGGGGTGAGTCACTG |
| wt LAC-f | CTGCCGGCCATGGAGAC |
| wt LAC-r | GCTGACCACCTCTTCAGACT |
| mEos LA/C-f | CATGCTGTTGCTCATTCTGGA |
| mEos LA/C-r | ACAAGTCCCCCTCCTTCTTGG |
| m*Lmna*-Frag1-f | TTGCATGCCTGCAGGTCGACAATCTCTTAAGAGTCCCAACTCAAGCCCA |
| m*Lmna*-Frag1-r | TAATCGCACTCATGGCCGGCAGGGTGACAGT |
| mEos-Frag2-f | GCCATGAGTGCGATTAAGCCAGACATGA |
| mEos-Frag2-r | GGTCTCGAATTCTCGTCTGGCATTGTCAGGCAAT |
| m*Lmna*-Frag3-f | ACGAGAATTCGAGACCCCGTCACAGCGG |
| m*Lmna*-Frag3-r | GTAGGCGGGTGATCCGAGTGGGCGA |
| m*Lmna*-Frag4-f | GGATCACCCGCCTACAAGAGAAGGAGGACCTGCAGGAGCT |
| m*Lmna*-Frag4-r | CTATGACCATGATTACGAATTCCCTAGGGCTGGAATCTGGTAAGGAA |

*Continued on next page*

*Continued*

| Primer | Sequence (5' - 3') |
|---|---|
| m*Lmna*-WT-f | GATCGATGTACACGGCCAGG |
| m*Lmna*-WT-r | GTCCTTCTGTCCAAGTCCCG |
| m*Lmna*-KI-f | GATCGATGTACACGGCCAGG |
| m*Lmna*-KI-r | GTGGACCACTGCATTGAGATTTT |
| *Lmna*-LR-5'-f | CCTAGGTTCCCTCCCCTAGAT |
| *Lmna*-LR-5'-r | GGTAACTTGACACCCTTCTCCTTAG |
| *Lmna*-LR-3'-f | GAGCATGCTGTTGCTCATTCT |
| *Lmna*-LR-3'-r | CCCATCTGTGCACATGACCT |
| T2A-f | GGCAAAAAAGAAAAAGGAATTCGGCAGTGGAGAGGGCAGA |
| T2A-r | TCACTGGGCCAGGATTCTCCTCGA |
| mCherry-f | AATCCTGGCCCAGTGAGCAAGGGCGAGGAGGAT |
| mCherry-r | AGCGAGCTCTAGTTAGAATTCCTTGTACAGCTCGTCCATGCCGCCGGT |
| hLaminA-NheI-f | GCGGCGGCTAGCATGGTGAGCAAGGGCGAGGAGC |
| hLaminA-NheI-r | GTATATACTAGTAGGAGCGGGTGACCAG |

## Statistical analysis

The effect size was estimated based on preliminary data or similar experiments performed in a different cell system and the software G*power (http://www.gpower.hhu.de/) was used to estimate the required sample size to achieve a minimum power of 0.8 (or higher, if experimentally feasible). Especially for predicted small effect sizes, large (>100) experimental sample sizes were chosen. Experiments were performed at least in triplicates. For statistical analysis, biological replicates (e.g. different clones of the same genotype) from all experiments were used and are stated in each figure legend as the number of n. If technical replicates (e.g. qPCR triplicates from the same sample) are displayed, it is stated in the figure legend. To summarize experimental data sets, average, standard deviation and standard error of the mean were calculated and displayed in bar graphs. In specific cases, box plots were chosen to display data to better visualize data distributions. In box plots the median was depicted within the first and third quartiles with the whiskers representing minimal and maximal datapoints excluding statistical outliers (according to the method of Tukey using the interquartile range). For statistical analysis, normal distribution of data was tested using the D'Agostino-Pearson and Shapiro Wilk normality test. Additionally, quantile-quantile plots were created and visually inspected for normal distribution. For normally distributed data sets, the two-tailed student's t test was used for statistical analysis (unpaired or paired, if data points were matched in pairs). The F test was used to determine whether variances of data sets are equal. Data expressed as proportions are not normally distributed and were transformed using the arcsin transformation (transformed value = acrsin $\sqrt{value}$). Ratios were transformed using the logarithmic transformation. If multiple comparisons were necessary (e.g. more than 2 data sets to compare with each other), ANOVA was used, including post-hoc tests for pairwise comparisons (Tukey). For non-normally distributed data, the Mann-Whitney U test was used, or, for multiple comparisons, the Kruskal-Wallis test.

## Acknowledgements

We are grateful to Harald Herrmann, DKFZ Heidelberg, Germany for the generous gift of recombinant lamin A and for his advice for lamin A in vitro assembly assays, and to Egon Ogris, Max Perutz Labs, Vienna, for the lamin A 3A6 antibody. We thank the Max Perutz Labs Biooptics facility for technical support with microscopic imaging and image analysis and technical support with FACS sorting. This study was funded by the Austrian Science Fund (FWF grant P26492-B20, P29713-B28 and P32512-B) to RF. KG is a recipient of a DOC Fellowship of the Austrian Academy of Sciences at the Max Perutz Labs, Medical University Vienna (ÖAW DOC 25725). NN was a recipient of an APART Fellowship of the Austrian Academy of Sciences at the Max Perutz Labs, Medical University Vienna

(APART 11657). YG and IB acknowledge financial support from the Israel Science Foundation (ISF) grant 1219/17 and from the S Grosskopf grant for 'Generalized dynamic measurements in live cells'. TD was a recipient of two COST Short Term Scientific Mission Fellowships (COST-STSM-BM1002-8698 and COST-STSM-BM1002-11436) and an EMBO short term fellowship (ASTF 316–2011).

## Additional information

### Funding

| Funder | Grant reference number | Author |
| --- | --- | --- |
| Austrian Science Fund | P26492-B20 | Roland Foisner |
| Austrian Academy of Sciences | APART 11657 | Nana Naetar |
| Israel Science Foundation | ISF grant 1219/17 | Irena Bronshtein<br>Yuval Garini |
| European Cooperation in Science and Technology | COST-STSM-BM1002-8698 | Thomas Dechat |
| European Molecular Biology Organization | ASTF 316-2011 | Thomas Dechat |
| S Grosskopf Grant | | Irena Bronshtein<br>Yuval Garini |
| Austrian Academy of Sciences | ÖAW DOC 25725 | Konstantina Georgiou |
| Austrian Science Fund | P29713-B28 | Roland Foisner |
| Austrian Science Fund | P32512-B | Roland Foisner |
| European Cooperation in Science and Technology | COST-STSM-BM1002-11436 | Thomas Dechat |

The funders had no role in study design, data collection and interpretation, or the decision to submit the work for publication.

### Author contributions

Nana Naetar, Conceptualization, Formal analysis, Supervision, Funding acquisition, Validation, Investigation, Writing - original draft, Writing - review and editing; Konstantina Georgiou, Irena Bronshtein, Formal analysis, Investigation, Methodology, Writing - original draft, Writing - review and editing; Christian Knapp, Software, Formal analysis, Investigation, Methodology, Writing - review and editing; Elisabeth Zier, Formal analysis, Investigation, Writing - review and editing; Petra Fichtinger, Resources, Investigation; Thomas Dechat, Conceptualization, Supervision, Writing - review and editing; Yuval Garini, Conceptualization, Resources, Supervision, Validation, Writing - review and editing; Roland Foisner, Conceptualization, Resources, Supervision, Funding acquisition, Validation, Project administration, Writing - review and editing

### Author ORCIDs

Nana Naetar  https://orcid.org/0000-0002-8978-466X
Christian Knapp  http://orcid.org/0000-0003-2463-5775
Thomas Dechat  https://orcid.org/0000-0003-3236-7889
Yuval Garini  https://orcid.org/0000-0002-8783-2015
Roland Foisner  https://orcid.org/0000-0003-4734-4647

### Decision letter and Author response

Decision letter https://doi.org/10.7554/eLife.63476.sa1
Author response https://doi.org/10.7554/eLife.63476.sa2

## Additional files

### Supplementary files

• Transparent reporting form

### Data availability

All data generated or analysed during this study are included in the manuscript and supporting files. Source data files have been provided for Figures 2G-H, Figure 4A-B and Figure 6A.

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

# Appendix 1

## Online supplemental material

*Figure 1—figure supplement 1* shows the characterization of HeLa LAP2α knockout clones created by CRISPR-Cas9 used in *Figure 1*. *Figure 1—videos 1–4* show live-cell imaging of HeLa wildtype and LAP2α knockout cells expressing either GFP-pre-lamin A (*Figure 1—videos 1* and *2*, corresponding to *Figure 1A*, panel 1 and 2, and *Figure 1—video 3* displaying an identically treated, second LAP2α knockout clone) or GFP-ΔK32 pre-lamin A (*Figure 1—video 4*, corresponding to *Figure 1A*, panel 3), imaged 5 hr post transfection. *Figure 1—videos 5–8* show live-cell imaging of HeLa wildtype and LAP2α knockout cells expressing either GFP-pre-lamin A (*Figure 1—videos 5* and *6*, corresponding to *Figure 1B*, panel 1 and 2, and *Figure 1—video 7* displaying an identically treated, second LAP2α knockout clone) or GFP-ΔK32 pre-lamin A (*Figure 1—video 8*, corresponding to *Figure 1B*, panel 3), imaged 24 hr post transfection. *Figure 1—figure supplement 2* shows live cell imaging of HeLa cells expressing mature GFP-lamin A or mature GFP-ΔK32 lamin A (associated with *Figure 1*). *Figure 1—videos 9–10* correspond to *Figure 1—figure supplement 2A* and *Figure 1—videos 11–12* correspond to *Figure 1—figure supplement 2B*. *Figure 2—figure supplement 1* shows the characterization of mEos3.2-lamin A/C mouse dermal fibroblasts used in *Figures 2–6*. *Figure 2—figure supplement 2* shows the characterization of isogenic mEos3.2-lamin A/C LAP2α knockout cell lines used in *Figures 2–6*. *Figure 2—source data 1* contains raw data for single cell measurements of DAPI and lamin A/C nucleoplasmic/peripheral ratio shown in *Figure 2G–H*. *Figure 2—figure supplement 3* compares the nucleoplasmic/peripheral signal ratios of lamin A/C and lamin B1 (associated with *Figure 2G–H*). *Figure 3—figure supplement 1* shows immunofluorescence images of various LAP2α knockout and wildtype mEos3.2 cells with different antibodies. *Figure 4—source data 1* contains raw data for constant photobleaching measurements of intranuclear lamin A/C depicted in *Figure 4A*. *Figure 4—source data 2* contains raw data for fluorescence correlation spectroscopy (FCS) measurements of intranuclear lamin A/C shown in *Figure 4B*. *Figure 4—figure supplement 1* shows lamin A/C extraction and lamin phosphorylation in wildtype and LAP2α knockout HeLa cells (associated with *Figures 4* and *5*). *Figure 6—figure supplement 1* shows characterization of *Lmna* knockout and *Lmna/Lap2α* double knockout mouse dermal fibroblasts used in *Figure 6*. *Figure 6—source data 1* contains raw data for measurements of telomere motion volume depicted in *Figure 6*.

