## [Decision Letter]

**Acceptance summary:**

This manuscript describes new tools to investigate the attributes specific to nucleoplasmic versus lamina-integrated A-type lamins. Based on their findings, the authors develop a new model in which LAP2α influences the conformational state of A-type lamins, antagonizing stable lamin A filament assembly with ramifications for chromatin mobility.

**Decision letter after peer review:**

Thank you for submitting your article "LAP2α maintains a mobile and low assembly state of A-type lamins in the nuclear interior" for consideration by *eLife*. Your article has been reviewed by three peer reviewers, including Megan C King as the Reviewing Editor and Reviewer #1, and the evaluation has been overseen by Suzanne Pfeffer as the Senior Editor.

The reviewers have discussed the reviews with one another and the Reviewing Editor has drafted this decision to help you prepare a revised submission.

Summary:

This work builds on prior studies by the Foisner group that investigated the function(s) of the soluble A-type lamin binding protein, LAP2α. One of their prior observations using antibody labeling was that there appeared to be a depletion of the nucleoplasmic pool of A-type lamins in cells lacking LAP2α. In this manuscript, the authors employ CRISPR-Cas9 editing to develop new tools to investigate the attributes specific to nucleoplasmic versus lamina-integrated A-type lamins. Using this new approach (and comparing it with their prior observations), the authors hit upon a new model in which LAP2α influences the conformational state of A-type lamins, which in turn influences its detection by a commonly used antibody. This technical detail explains the new realization that nucleoplasmic lamin A persists in LAP2α-null cells, albeit in a different state. The authors provide evidence that LAP2α antagonizes stable lamin A filament assembly, that is absence leads to stabilized intranuclear lamin A assemblies, and that telomere mobility is negatively influenced by loss of LAP2α in a manner depending on the presence of lamin A/C. The authors' work further identifies two pathways by which nucleoplasmic lamins emerge, namely by 1) initial localization to the lamina followed by relocalization to the nucleoplasm, and 2) from the pool of mitotic lamins which are not associated to the lamina.

Overall there was enthusiasm for the study, with the reviewers stating their appreciation for the author's mechanistic approach to studying lamin assembly state and the use of complementary cell biology/microscopy and biochemical approaches. The rigor of the science was also lauded, including inclusion of, for example, genome editing quality control measures. Taken together the reviewers felt that the findings provided a new perspective on how LAP2α influences the state of A-type lamins. As the impact of lamins on nuclear organization is critical for nuclear functions and important for nuclear integrity, these results are fundamental for the understanding of both lamin A/C and LAP2α. The reviewers also made specific suggestions for a small number of experimental revisions as well as further editing of the text to improve clarity and to provide greater context.

Essential revisions:

1) Structure of the Results. Given that a major focus of the paper is to explain conflicting results with (the same group's) prior published data on the effect of LAP2α depletion, it would have helped to lay this out more clearly from the outset of the paper. As written, the reader is confused until arriving at Figure 3. While the reviewers appreciate that resolving this conflict leads to a new perspective – namely that LAP2α influences the state of the lamin assembly in a way that disrupts its detection by the N18 antibody, structuring the manuscript to get to this point as quickly as possible would improve its accessibility.

2) Suggested edits to clarify the manuscript. Although the reviewers appreciated the transparency of the authors in demonstrating their workflow and quality control measures, some of the terminology made the manuscript difficult to read. For example, the manuscript would be easier to understand if cell lines were given descriptive names (e.g.: LAP2α KO, or mEos3.2-*lmna* instead of "WT#21") rather than continuing to refer to them by the small guide RNA that was used to generate them. A second example: while presenting biological replicate data as in Figure 1 is appreciated, it was challenging to ascertain that the second and third columns in panels A and B were biological replicates. The authors could improve clarity by presenting one biological replicate in the main text with additional replicate(s) moved to the supplement, especially considering that it appears that only one of the clones was used for the quantifications shown in the bottom panels.

3) Alternative explanations. Further comment and/or clarification is necessary when discussing the state of nucleoplasmic lamin polymerization and how the dynamic measurements reflect lamin assembly state versus anchorage to other lamin binding proteins. For example, although the nuclear lamina filaments are typically 200-400 nm in length, they are also very flexible. A 200 nm filament would have a molecular weight of <1.4MDa ( ~50% of a ribosome) and can be bent and curved. As a consequence, a single filament would be expected to have a reasonably high diffusion coefficient. At the lamina, lamins are less mobile, however, it is likely to be due to binding partners that anchor lamins to the INM and chromatin rather than assembly state as the diffusion of even a 1.4 MDa protein complexes is quite fast. Thus, the authors need to be careful in their interpretation as nucleoplasmic lamins may be polymerized but more mobile (less anchored) than their lamina-bound lamin population without a change in assembly state. Along these lines, greater explanation for the apparent paradox between the increase in immobile fraction by FRAP and the increased diffusion coefficient by FCS in the LAP2α-depleted condition is needed. The authors suggest that the latter is due to the loss of LAP2α binding (subsection “LAP2α loss changes properties of nucleoplasmic lamin A/C rendering them less mobile and more resistant to biochemical extraction”), but some modeling would go a long way here. What form are the lamins thought to be in, and how does the bulk that LAP2α would bring match the apparent changes in diffusivity?

4) Edited clones. It was noted that the authors chose to disrupt LAP2α at the beginning of exon 4, but the reason for this was unclear. For the readers' clarity, can the authors include the size of exons 1 and 2 in the schematic in the supplementary figures? Can the authors provide evidence that the "LAP2α KO" cells used in the experiments do not express a truncated LAP2α protein that could influence the results and interpretation?

5) Concerns over whether lamin tagging may influence the results. The authors make no comment on the functionality of the mEos-tagged lamin A/C CRISPR lines. However, the comment suggesting that some clones could have altered nuclear morphology (subsection “LAP2α-deficiency does neither affect the formation nor the steady state level of endogenous lamin A in the nuclear interior”) raises some questions. How did the authors interpret this? Were these clones in which there were indels in some *lmna* alleles affecting the levels? Or is this a consequence of the fusion? One approach we suggest is to compare morphology and overall fitness of cells with all untagged *lmna* with cells with all tagged *lmna*, to determine whether the tagged proteins are fully functional. How do the authors explain the relatively low expression level of the mEos fusion relative to the untagged? If the MDFs are diploid, presumably we would expect this to be one allele tagged and one allele untagged. Given that the expression ratio is very different from this, could the tagged lamin A/C be targeted for degradation? As these cell lines are critical for the rest of the study, this information is important, particularly as many lamin mutations do not cause obvious phenotypes in tissue culture cells, but defects can still emerge during development and aging in the context of an animal. Further, in the subsection “LAP2α loss changes properties of nucleoplasmic lamin A/C rendering them less mobile and more resistant to biochemical extraction”, the authors describe a set of experiments that are meant to demonstrate that their failure to see a difference in nucleoplasmic A-type lamins in LAP2α mutants is not due to the fluorescent protein tag used, however, instead of looking at untagged lamins, they elect to look at a cell line that has all *lmna* alleles tagged. The authors' statements would be strengthened if the LAP2α KO cells from Figure 1 were stained with both the 3A6 antibody and the N18 antibody to determine whether untagged lamins behave the same way as tagged lamins. More generally, including experiments that allow the reader to compare directly between a cell line with all *lmna* alleles tagged and a cell line with no *lmna* alleles tagged is important.

6) Implications for regulation of A-type lamin assembly. The authors build a convincing case that binding to A-type lamins by LAP2α influences their ability to assemble. But how do cells leverage this relationship for biological functions? Do cells tune the amount of fully soluble vs. partially assembled A-type lamins in the nucleoplasm in order to control nuclear structure or function in response to certain stimuli (e.g. cyclical stretch, for example)? Have the A-type lamins in the nucleoplasm been found to be in a different assembly state in different cell types? Similarly, one prediction that arises from the proposed model is that regulation of LAP2α levels will modulate the relative pool of A-type lamins at the nuclear interior versus the nucleoplasm. Beyond the knock-out cells, is there any other evidence of this relationship? How do the authors think the membrane integrated LAP2β fits into the story? Any further insight provided would greatly help to address biological mechanism.

7) Lamin assembly mechanism. The authors show that nucleoplasmic lamins are first localized to the lamina, where they can polymerize. Isn't it possible that filaments can be released into the nucleoplasm? The authors suggest that *Lap2α* keeps lamin in a less polymerized state based in part on the findings that *Lap2α* inhibits lamin A assembly in vitro. However, previous work by Zwerger et al., 2015, showed that inhibitors of in vitro lamin A assembly have no impact on incorporation and localization of lamin A into the lamina, while incorporation of lamin A into the nuclear lamina was abolished when other lamin binders that have no effect on lamin assembly in vitro were used. This suggests that either in vitro assembly is not representing the cellular lamin assembly or assembly of lamin into the lamina is independent of polymerization states of lamins. The authors should discuss these views in the revised manuscript. Greater use of the ΔK32 mutant was also suggested, specifically as related to Figure 1, where the data on this assembly mutant should be included in the primary figures and with regards to the sensitivity of the N18 antibody – does this detect the ΔK32 mutated lamin A? Could this provide further insight into the impact of LAP2α by extension?

8) Chromatin dynamics. Over what time interval was the volume of movement for the telomeres observed? This is important because more fluctuations in nuclear position, for example, could influence this measure. In addition, telomeres are a confusing choice, given abundant evidence that there is crosstalk between the state of the nuclear lamina and telomere biology (e.g. lamin mutants affecting telomere homeostasis, etc.). At a minimum, acknowledging that telomeres may not reflect the effect on chromatin globally is important. Examples of the raw mean squared displacements would be more informative. Is the difference between *lmna* KO and *lmna/Lap2α* DKO (Figure 6 right panel) significant?

---

## [Author Response]

Essential revisions:1) Structure of the Results. Given that a major focus of the paper is to explain conflicting results with (the same group's) prior published data on the effect of LAP2α depletion, it would have helped to lay this out more clearly from the outset of the paper. As written, the reader is confused until arriving at Figure 3. While the reviewers appreciate that resolving this conflict leads to a new perspective – namely that LAP2α influences the state of the lamin assembly in a way that disrupts its detection by the N18 antibody, structuring the manuscript to get to this point as quickly as possible would improve its accessibility.

We understand the concerns of the reviewers and have now rewritten the Abstract and the last paragraph in the Introduction to more clearly point out the aims of this study and the surprising finding that led to a new adapted model of LAP2α function in the regulation of nucleoplasmic lamins. In addition, we streamlined the text describing the formation of the nucleoplasmic lamin pool in the Results (Figures 1 and 2) and changed the focus more towards the effect of LAP2α rather than the pathways involved in the formation of lamins A/C in the nuclear interior.

We also considered rearranging the figures, but after extensive discussion we prefer to keep the current order of figures, as rearrangements would make the structure of the manuscript less clear and disturb the logical flow of the text and experimental set up (e.g. ectopic proteins versus endogenously labeled lamin A/C). We are confident that with the included text changes we can best address the reviewer’s comments, while maintaining the order of the figures.

2) Suggested edits to clarify the manuscript. Although the reviewers appreciated the transparency of the authors in demonstrating their workflow and quality control measures, some of the terminology made the manuscript difficult to read. For example, the manuscript would be easier to understand if cell lines were given descriptive names (e.g.: LAP2α KO, or mEos3.2-lmna instead of "WT#21") rather than continuing to refer to them by the small guide RNA that was used to generate them. A second example: while presenting biological replicate data as in Figure 1 is appreciated, it was challenging to ascertain that the second and third columns in panels A and B were biological replicates. The authors could improve clarity by presenting one biological replicate in the main text with additional replicate(s) moved to the supplement, especially considering that it appears that only one of the clones was used for the quantifications shown in the bottom panels.

We now use descriptive names for the cell lines throughout the text and in most figures. In figures, where different LAP2α knockout clones generated by different sgRNAs were used, as well as cell clones treated with the same sgRNAs, but still expressing LAP2α, we designated the cells LAP2α KO sgxx and WT sgxx ctrl (see for example Figure 2F and G). These designations are clearly explained in the figure legend of Figure 2. We agree that these changes make it much easier for a broad readership to follow the logical flow of the manuscript.

In order to address the point of the biological replicate, we removed replicate panels in Figure 1A and B. Instead of showing replicate panels in Figure 1, we now include these data in additional video files (Videos 3 and 7).

3) Alternative explanations. Further comment and/or clarification is necessary when discussing the state of nucleoplasmic lamin polymerization and how the dynamic measurements reflect lamin assembly state versus anchorage to other lamin binding proteins. For example, although the nuclear lamina filaments are typically 200-400 nm in length, they are also very flexible. A 200 nm filament would have a molecular weight of <1.4MDa ( ~50% of a ribosome) and can be bent and curved. As a consequence, a single filament would be expected to have a reasonably high diffusion coefficient. At the lamina, lamins are less mobile, however, it is likely to be due to binding partners that anchor lamins to the INM and chromatin rather than assembly state as the diffusion of even a 1.4 MDa protein complexes is quite fast. Thus, the authors need to be careful in their interpretation as nucleoplasmic lamins may be polymerized but more mobile (less anchored) than their lamina-bound lamin population without a change in assembly state. Along these lines, greater explanation for the apparent paradox between the increase in immobile fraction by FRAP and the increased diffusion coefficient by FCS in the LAP2α-depleted condition is needed. The authors suggest that the latter is due to the loss of LAP2α binding (subsection “LAP2α loss changes properties of nucleoplasmic lamin A/C rendering them less mobile and more resistant to biochemical extraction”), but some modeling would go a long way here. What form are the lamins thought to be in, and how does the bulk that LAP2α would bring match the apparent changes in diffusivity?

We toned down the conclusions on lamin polymerization and included alternative explanations based on lamin filament anchorage to explain the observed differences in lamin A mobility (i.e. movement) in LAP2α KO versus WT cells throughout the text and added a paragraph on this topic in the Discussion.

The increase in the immobile fraction in the absence of LAP2α as measured by constant photobleaching suggests that a larger fraction of intranuclear lamins are immobile, likely because they are bound to an entity that prevents its motion (e.g. chromatin or larger lamin filament assemblies in the nucleoplasm). These immobile structures cannot be measured by FCS, as their fluorescence intensity is not fluctuating. Hence, FCS data refer exclusively to the unbound, mobile fraction of lamin A, which is reduced (but still present) in LAP2α knockout cells. This mobile lamin A/C fraction may be diffusing faster as the complexes without LAP2α are smaller than in WT cells. In the manuscript we explain the results more clearly and give an estimation of a potential correlation of diffusion coefficient changes with changes in the molecular mass of the remaining dynamic lamin complex in the absence of LAP2α. The exact form these mobile lamin-LAP2α complexes are in is unknown. For the calculations, we are assuming the simplest form with one molecule each present in the complexes. Notably, our unpublished data suggest a stoichiometry of 1:1 or 1:2 for soluble lamin A/C-LAP2α complexes.

4) Edited clones. It was noted that the authors chose to disrupt LAP2α at the beginning of exon 4, but the reason for this was unclear. For the readers' clarity, can the authors include the size of exons 1 and 2 in the schematic in the supplementary figures? Can the authors provide evidence that the "LAP2α KO" cells used in the experiments do not express a truncated LAP2α protein that could influence the results and interpretation?

We now clarify in the figure legend that exon 4 is the first and only LAPα-specific exon in the LAP2 gene, while exons 1-3 are common to all LAP2 isoforms. Thus, generating a LAP2α-specific knockout without disturbing expression of the other isoforms requires targeting the beginning of exon 4. To address the reviewer’s specific point, we also added exons 1 to 3 in the schematic drawings of Figure 1—figure supplement 1 and Figure 2—figure supplement 2.

As for a truncated LAP2α fragment, one LAP2α KO clone (sg3) indeed expresses low levels of a small LAP2α fragment (see Figure 2—figure supplement 2D), but we mostly used sg1 clones for the analyses, which do not contain a fragment (Figure 2—figure supplement 2D). In experiments, where sg3 was used (data in Figure 2E-G, Figure 3—figure supplement 1A), the respective data are clearly labeled and explained in the legend, and we included a heterozygous clone that was treated with sg3 (former clone sg3-2, now labeled with WT sg3 ctrl) as control, expressing LAP2α from 1 allele and a short truncated LAP2α fragment from another targeted allele (see Figure 2—figure supplement 2C and D). WT sg3 ctrl expresses full length LAP2α at WT levels (Figure 2B).

5) Concerns over whether lamin tagging may influence the results. The authors make no comment on the functionality of the mEos-tagged lamin A/C CRISPR lines. However, the comment suggesting that some clones could have altered nuclear morphology (subsection “LAP2α-deficiency does neither affect the formation nor the steady state level of endogenous lamin A in the nuclear interior”) raises some questions. How did the authors interpret this? Were these clones in which there were indels in some lmnA alleles affecting the levels? Or is this a consequence of the fusion? One approach we suggest is to compare morphology and overall fitness of cells with all untagged lmna with cells with all tagged lmna, to determine whether the tagged proteins are fully functional. How do the authors explain the relatively low expression level of the mEos fusion relative to the untagged? If the MDFs are diploid, presumably we would expect this to be one allele tagged and one allele untagged. Given that the expression ratio is very different from this, could the tagged lamin A/C be targeted for degradation? As these cell lines are critical for the rest of the study, this information is important, particularly as many lamin mutations do not cause obvious phenotypes in tissue culture cells, but defects can still emerge during development and aging in the context of an animal. Further, in the subsection “LAP2α loss changes properties of nucleoplasmic lamin A/C rendering them less mobile and more resistant to biochemical extraction”, the authors describe a set of experiments that are meant to demonstrate that their failure to see a difference in nucleoplasmic A-type lamins in LAP2α mutants is not due to the fluorescent protein tag used, however, instead of looking at untagged lamins, they elect to look at a cell line that has all lmna alleles tagged. The authors' statements would be strengthened if the LAP2α KO cells from Figure 1 were stained with both the 3A6 antibody and the N18 antibody to determine whether untagged lamins behave the same way as tagged lamins. More generally, including experiments that allow the reader to compare directly between a cell line with all lmna alleles tagged and a cell line with no lmna alleles tagged is important.

The clone used in most experiments (mEos3.2-*Lmna* WT clone #21, see Figure 2—figure supplement 1D and F) is tetraploid (as determined from sequencing data and TIDE analysis, see Figure 2—figure supplement 2B and C) and expresses tagged lamin A/C from 1 allele and untagged lamin A/C from 3 alleles (see Figure 2—figure supplement 1G for ratio of tagged versus untagged lamin A mRNA and protein levels in newly added protein quantification).

There may be a certain influence of the tag on protein stability, as the ratio of tagged to untagged lamin A/C protein is shifted towards approx. 1:8 on the protein level (Figure 2—figure supplement 1G), but functions and properties of tagged lamin A and C seem to be unaffected. In the manuscript, we refer to the correct localization of tagged lamin A (Figure 2C), to the stable integration into the lamina by FRAP (former Figure 2—figure supplement 1H, now moved to Figure 2 as new panel D), normal post-mitotic lamina assembly (shown in Figure 2) and similar biochemical properties, such as solubility in salt/detergent buffer (newly added quantification of solubility in Figure 2—figure supplement 1G, right panel) (subsection “LAP2α-deficiency does neither affect the formation nor the steady state level of endogenous lamin A in the nuclear interior”).

However, while some mEos3.2-*Lmna* clones (independent of whether WT lamin A/C was still present or not) had altered nuclear morphology, this often correlated with very low total lamin A/C levels due to indels in some *Lmna* alleles as also pointed out by the reviewers. We chose two clones expressing only mEos-tagged lamin A/C, one WT and one KO for LAP2α (WT#5 and LAP2α KO#17, see Figure 2—figure supplement 1D and F) for further control experiments, as suggested by the reviewers. The WT clone expressing only tagged lamin A/C has lower total levels of lamin A/C (likely expressed only from 1 allele while other alleles contain indels), but still displays normal nuclear morphology (see Figure 2—figure supplement 1I). Following the suggestion of the reviewers to test the fitness of these cell clones and to confirm that tagged lamin A/C behaves like untagged lamins, we added additional data comparing the parental WT and LAP2α KO cells (no tagged lamins) with the cell clones expressing only tagged lamin A/C regarding cell viability, nuclear morphology, lamin localization and lamina structure (Figure 2—figure supplement 1H and I). Notably, WT#5 cells expressing lower total levels of mEos-tagged lamins A/C displays a slight reduction in cell viability (pointed out in the figure legend), which is likely caused by reduced total lamin levels rather than the tagging of lamins, as LAP2α KO#17 cells expressing higher mEos-lamin A/C levels display normal viability, when compared to parental cells with untagged *Lmna* alleles (Figure 2—figure supplement 1H).

Additionally, we performed immunofluorescence microscopic analyses of these cells using anti-lamin A/C N18 and 3A6 antibody and confirmed the lack of intranuclear lamin staining in the absence of LAP2α in both, cells without tagged lamins and cell clones expressing only tagged lamins (Figure 3—figure supplement 1C and D). Thus, by all these means cells with only tagged lamins behave as cells with untagged lamins.

We also discuss functionality of tagged lamin A/C now in more detail in the Results section.

6) Implications for regulation of A-type lamin assembly. The authors build a convincing case that binding to A-type lamins by LAP2α influences their ability to assemble. But how do cells leverage this relationship for biological functions? Do cells tune the amount of fully soluble vs. partially assembled A-type lamins in the nucleoplasm in order to control nuclear structure or function in response to certain stimuli (e.g. cyclical stretch, for example)? Have the A-type lamins in the nucleoplasm been found to be in a different assembly state in different cell types? Similarly, one prediction that arises from the proposed model is that regulation of LAP2α levels will modulate the relative pool of A-type lamins at the nuclear interior versus the nucleoplasm. Beyond the knock-out cells, is there any other evidence of this relationship? How do the authors think the membrane integrated LAP2β fits into the story? Any further insight provided would greatly help to address biological mechanism.

In the revised manuscript, we discuss a potential biological function in great detail.

As we have shown previously, LAP2α loss clearly impairs the detection of nucleoplasmic lamins by antibodies in various cell types, including fibroblasts and epidermal and colon progenitor cells (Naetar et al., 2008). Changes in antibody-stained lamin A/C levels in the nuclear interior can also be seen in physiological conditions within cell populations in a cell culture, expressing different levels of LAP2α (see e.g. Figure 1—figure supplement 1E, depicting cells with different levels of LAP2α side by side).

As for a potential biological function and regulation of nucleoplasmic lamins we discuss the following points: While the response of lamin A solubility to mechanical stretch works via lamin A phosphorylation (Buxboim et al., 2014), changes in the detectability of lamin A/C in the nuclear interior during differentiation may be regulated by differentiation-specific downregulation of LAP2α (Markiewicz, Ledran, and Hutchison, 2005; Naetar et al., 2007; Pekovic et al., 2007). Similar effects can be seen in progeria premature aging disease (Vidak et al., 2018; Vidak et al., 2015). We hypothesize that a mobile nucleoplasmic lamin A is important for chromatin regulation and gene expression [see (Bronshtein et al., 2015; Gesson et al., 2016), our unpublished data, and shown in this manuscript in Figure 6], as well as for the regulation of retinoblastoma protein (Dorner et al., 2006) during cell cycle exit and initiation of cell differentiation.

LAP2β was shown to bind lamin B, not lamin A/C (Foisner and Gerace, 1993), while LAP2α binds lamin A/C via a LAP2α-specific domain encoded by exon 4 (Dechat et al., 2000) absent in LAP2β. Thus, we feel that the function of LAP2α regulating lamin A/C in the nuclear interior is unique to the protein and not affected by the other LAP2 isoforms.

7) Lamin assembly mechanism. The authors show that nucleoplasmic lamins are first localized to the lamina, where they can polymerize. Isn't it possible that filaments can be released into the nucleoplasm? The authors suggest that Lap2α keeps lamin in a less polymerized state based in part on the findings that Lap2α inhibits lamin A assembly in vitro. However, previous work by Zwerger et al., 2015, showed that inhibitors of in vitro lamin A assembly have no impact on incorporation and localization of lamin A into the lamina, while incorporation of lamin A into the nuclear lamina was abolished when other lamin binders that have no effect on lamin assembly in vitro were used. This suggests that either in vitro assembly is not representing the cellular lamin assembly or assembly of lamin into the lamina is independent of polymerization states of lamins. The authors should discuss these views in the revised manuscript. Greater use of the ΔK32 mutant was also suggested, specifically as related to Figure 1, where the data on this assembly mutant should be included in the primary figures and with regards to the sensitivity of the N18 antibody – does this detect the ΔK32 mutated lamin A? Could this provide further insight into the impact of LAP2α by extension?

We discuss now in great detail that lamin A assembly may be different in vivo versus in vitro, in view of the data shown in Zwerger et al., 2015, and we also describe alternative models on the assembly state and interactions of nucleoplasmic lamins (see also our response to point 3 above) (Discussion).

In order to highlight the results of the assembly-deficient ΔK32 prelamin A mutant we moved panels showing assembly of this lamin mutant from Figure 1—figure supplement 2 to Figure 1.

Stainings of ΔK32 lamin A mutant by the anti-lamin A/C N18 antibody in LAP2α WT versus KO cells, as mentioned by the reviewer, were already done and shown in one of our previous papers (Pilat et al., 2013). As expected, unlike WT lamin A, the intranuclear staining of the assembly-deficient ΔK32 lamin A by antibody N18 was not impaired in LAP2α KO cells, which would be in support of lamin A assembly being responsible for loss of N18 staining. We add these findings in the Discussion.

8) Chromatin dynamics. Over what time interval was the volume of movement for the telomeres observed? This is important because more fluctuations in nuclear position, for example, could influence this measure. In addition, telomeres are a confusing choice, given abundant evidence that there is crosstalk between the state of the nuclear lamina and telomere biology (e.g. lamin mutants affecting telomere homeostasis, etc.). At a minimum, acknowledging that telomeres may not reflect the effect on chromatin globally is important. Examples of the raw mean squared displacements would be more informative. Is the difference between lmna KO and lmna/Lap2α DKO (Figure 6 right panel) significant?

Each nucleus is measured for 20.5 minutes. Our extensive experience on measurements in this time range has shown that this is long enough to prevent that measurements are not affected by short-time motion, which may not represent the actual dynamics. [see also (Zada et al., 2019)].

Furthermore, the motion of the nucleus itself (“nuclear position”) is completely controlled during the analysis by performing a drift-correction algorithm to each measured cell. We use a robust algorithm (see for example: (Bronshtein et al., 2015), section methods; and in “Single particle tracking for studying the dynamic properties of genomic regions in live cells“ , I. Bronshtein Berger, E. Kepten and Y. Garini, In: Imaging gene expression Methods and Protocols, edited by Y. Shav-Tal, Springer 2013). Due to the fact that we can track simultaneously (over 20.5 minutes) at least 20-40 telomeres in each cell, and because they are spread in the nucleus volume, the drift-correction algorithm leads to excellent correction of the cell motion. This is done by calculating for each image (at each time point) the “center of gravity” of the nucleus and shifting the coordinates of the nucleus to the same very point. Only then we calculate the dynamics of each telomere, and therefore, the cell motion is not affecting the calculated track of the telomeres and the calculations of the MSD. In addition, we also perform a rotation-correction of each nucleus by using an adequate algorithm that is part of the routine analysis. These two corrections ensure that the MSD that we calculate is accurate and it is not affected by cell drift or rotation.

Regarding the choice of telomeres, we have performed numerous measurements of the dynamics of telomeres in different cell types (MEFs, U2OS and HeLa) while depleting different proteins, and both telomeres and centromeres in WT and lamin A-depleted U20S cells. It is our experience that the dynamics of both types of probes in all these cells behave in a similar manner, i.e. if the telomere MSD reduces, also the centromere MSD reduces and vice versa (Bronshtein et al., 2015). Therefore, we strongly believe that the effect that we measure is not related to the biological properties of the telomeres themselves, and they only serve as ‘beacons’ for observing the general dynamics of chromatin in the nucleus. Possible modifications to the telomeres biology itself that may result from protein binding or other modifications, may change the dynamics of the telomeres on the scale of sub-telomere size, most probably in a 10-20 nm motion range, while the volume of motion that we measure is roughly a sphere with a diameter of ~250-300 nm. Local internal-dynamics of the telomere cannot affect the volume of its motion as a whole, which is what we are measuring.

As requested, for clarifying the data to the readers, we are adding a graph that shows the MSD of telomeres in WT and LAP2α knockout cells to Figure 6 (new panel B). The graph clearly shows that the absence of LAP2α leads to a ‘slower’, more constraint diffusion with lower MSD. These data are correlated with the change in the volume of motion shown in Figure 6A.

We also calculated the corrected p value for comparing *Lmna* KO and *Lmna/Lap2α* DKO, which is highly significant (Dunn's multiple comparisons test p < 0.0001 as part of Kruskal-Wallis post tests). Despite the rather small effect size (Cohen’s d=0.3), this suggests that LAP2α has a small, but significant effect on chromatin motion independent of lamin A/C, where its presence further restricts free diffusion of chromatin. This is indicated by the increase in chromatin motion volume in DKO compared to *Lmna* single KO cells (Figure 6A). These observations were included in the Results and Discussion sections of the manuscript.

References:

Foisner, R., and Gerace, L. (1993). Integral membrane proteins of the nuclear envelope interact with lamins and chromosomes, and binding is modulated by mitotic phosphorylation. Cell, 73(7), 1267-1279. doi:10.1016/0092-8674(93)90355-tPekovic, V., Harborth, J., Broers, J. L., Ramaekers, F. C., van Engelen, B., Lammens, M., von Zglinicki, T., Foisner, R., Hutchison, C., and Markiewicz, E. (2007). Nucleoplasmic LAP2alpha-lamin A complexes are required to maintain a proliferative state in human fibroblasts. J Cell Biol, 176(2), 163-172. doi:10.1083/jcb.200606139Vidak, S., Georgiou, K., Fichtinger, P., Naetar, N., Dechat, T., and Foisner, R. (2018). Nucleoplasmic lamins define growth-regulating functions of lamina-associated polypeptide 2alpha in progeria cells. J Cell Sci, 131(3). doi:10.1242/jcs.208462Zada, D., Bronshtein, I., Lerer-Goldshtein, T., Garini, Y., and Appelbaum, L. (2019). Sleep increases chromosome dynamics to enable reduction of accumulating DNA damage in single neurons. Nat Commun, 10(1), 895. doi:10.1038/s41467-019-08806-w